# Defining the role of pulmonary endothelial cell heterogeneity in the response to acute lung injury

Terren K Niethamer[1,2,3,4], Collin T Stabler[1,2,3,4], John P Leach[1,2,3,4], Jarod A Zepp[1,2,3,4], Michael P Morley[1,3,4], Apoorva Babu[1,4], Su Zhou[1,4], Edward E Morrisey[1,2,3,4,5]*

[1]Department of Medicine, University of Pennsylvania, Philadelphia, United States; [2]Department of Cell and Developmental Biology, University of Pennsylvania, Philadelphia, United States; [3]Penn-Children's Hospital of Philadelphia Lung Biology Institute, University of Pennsylvania, Philadelphia, United States; [4]Penn Cardiovascular Institute, University of Pennsylvania, Philadelphia, United States; [5]Penn Institute for Regenerative Medicine, University of Pennsylvania, Philadelphia, United States

**Abstract** Pulmonary endothelial cells (ECs) are an essential component of the gas exchange machinery of the lung alveolus. Despite this, the extent and function of lung EC heterogeneity remains incompletely understood. Using single-cell analytics, we identify multiple EC populations in the mouse lung, including macrovascular endothelium (maEC), microvascular endothelium (miECs), and a new population we have termed *Car4*-high ECs. *Car4*-high ECs express a unique gene signature, and ligand-receptor analysis indicates they are primed to receive reparative signals from alveolar type I cells. After acute lung injury, they are preferentially localized in regenerating regions of the alveolus. Influenza infection reveals the emergence of a population of highly proliferative ECs that likely arise from multiple miEC populations and contribute to alveolar revascularization after injury. These studies map EC heterogeneity in the adult lung and characterize the response of novel EC subpopulations required for tissue regeneration after acute lung injury.

*For correspondence: emorrise@pennmedicine.upenn.edu

## Introduction

The primary function of the respiratory system is to exchange oxygen and other gases with the external environment. The functional compartment or niche responsible for gas exchange in the lungs is called the alveolus and is comprised of multiple epithelial, endothelial, and mesenchymal cell lineages. The pulmonary vascular plexus interacts intimately with alveolar type 1 (AT1) cells to form a very thin, gas-diffusible interface. The lung alveolus and its gas-exchange function is seriously disrupted upon acute lung injury such as occurs during a viral infection (*Gregory and Kobzik, 2015*; *Herold et al., 2015*; *Kash and Taubenberger, 2015*; *Kumar et al., 2011*; *Pociask et al., 2017*; *Zacharias et al., 2018*). Despite the functional importance of the lung alveolus, only recently have studies begun to elucidate its overall cellular complexity (*Cohen et al., 2018*; *Ding et al., 2018*; *He et al., 2018*; *Montoro et al., 2018*; *Plasschaert et al., 2018*; *Reyfman et al., 2019*; *Sabbagh et al., 2018*; *Treutlein et al., 2014*; *Xie et al., 2018*; *Xu et al., 2016*; *Zacharias et al., 2018*; *Zepp et al., 2017*). An improved understanding of the molecular mechanisms of alveolar regeneration is critical for the development of targeted therapies that induce or improve this process to restore functional gas exchange in pulmonary disease or following acute injury to the lung.

Previous work has demonstrated that acute or serious lung injury can lead to the generation of dense epithelial remodeling resulting in hypoxic vasoconstriction in regions of severe damage

**eLife digest** Animal lungs are filled with tiny air sacks called alveoli, where the gas exchanges that keep organisms alive can take place. Small blood vessels known as capillaries come in close contact with the alveoli, allowing oxygen to be extracted from the air into the blood, and carbon dioxide to be released from the blood into the air. The cells that line the inside of these capillaries (known as pulmonary endothelial cells) are important actors in these exchanges.

After having been damaged, for example by viruses like influenza, the lungs need to regenerate and create new capillaries. Yet, it was still unclear how pulmonary endothelial cells participate in the healing process, and if capillaries contain several populations of endothelial cells that play different roles.

To investigate this question, Niethamer et al. used an approach called single-cell analytics to examine individual endothelial cells in the alveoli of mice infected with influenza. This revealed that different subtypes of endothelial cells exist in capillaries, and that some may be able to perform slightly different jobs during lung recovery.

Niethamer et al. found that all subtypes could quickly multiply after injury to create more endothelial cells and re-establish gas exchanges. However, one newly identified group (called Car4-high ECs) was particularly primed to receive orders from damaged alveoli. These cells were also often found at the sites where the alveoli were most injured.

Lung injuries are a major cause of death worldwide. Understanding how pulmonary endothelial cells work when the organ is both healthy and injured should help to find ways to boost repair, and to create therapies that could target these cells.

(*Xi et al., 2017*). Proper tissue regeneration in the lung requires neovascularization of the alveolar region, and previous work has suggested that certain EC populations may be primed for proliferation when cultured in vitro (*Alvarez et al., 2008*; *Lee et al., 2017*; *Taha et al., 2017*). Cell surface markers such as CD34 and CD309 are enriched on mouse ECs that proliferate in culture (*Alvarez et al., 2008*; *Asahara et al., 1997*). Putative EC progenitors can be isolated from the peripheral blood or the liver and transplanted into injured regions to contribute to regeneration (*Asahara et al., 1997*; *Wakabayashi et al., 2018*). However, whether such EC subpopulations contribute to in vivo tissue homeostasis and response to injury in the adult lung remains unknown.

To address these questions and to define pulmonary EC heterogeneity at homeostasis and during regeneration, we utilized single cell RNA sequencing (scRNA-seq) analysis of the adult mouse lung, both uninjured and after acute influenza-induced viral injury. In addition to identifying microvascular (miEC) and both arterial and venous macrovascular (maEC) populations, we identified a new population we have termed *Car4*-high ECs that possess a unique transcriptome. *Car4*-high ECs express high levels of *Car4* and *Cd34*, are found throughout the lung periphery at homeostasis, and are primed to respond to Vegfa signaling based on their high expression of Vegf receptors, which corresponds to a receptor-ligand interaction analysis between *Car4*-high ECs and AT1 cells, their epithelial co-partners in gas exchange. *Car4*-high ECs are enriched in the regenerating zones surrounding the most damaged regions of the lung following influenza- or bleomycin-induced lung injury during the subsequent tissue regeneration process. Influenza injury revealed the emergence of a unique population of highly proliferative ECs, which are closely related to *Car4*-low miECs in gene expression but likely arise from multiple EC populations to contribute to regeneration of the pulmonary vasculature after injury by replacing lost endothelium. These data provide critical insight into lung EC heterogeneity and the key roles that differing subtypes of pulmonary ECs play in the response to acute lung injury.

## Results

### Pulmonary endothelial heterogeneity in the adult mouse lung

To define the heterogeneity of the adult pulmonary endothelium, we employed scRNA-seq analysis. We used the pan-endothelial marker CD31 (*Pecam1*) to isolate ECs (CD45⁻/CD31⁺) by FACS and sequenced using the 10X Genomics platform (*Figure 1A*). We identified seven separate clusters,

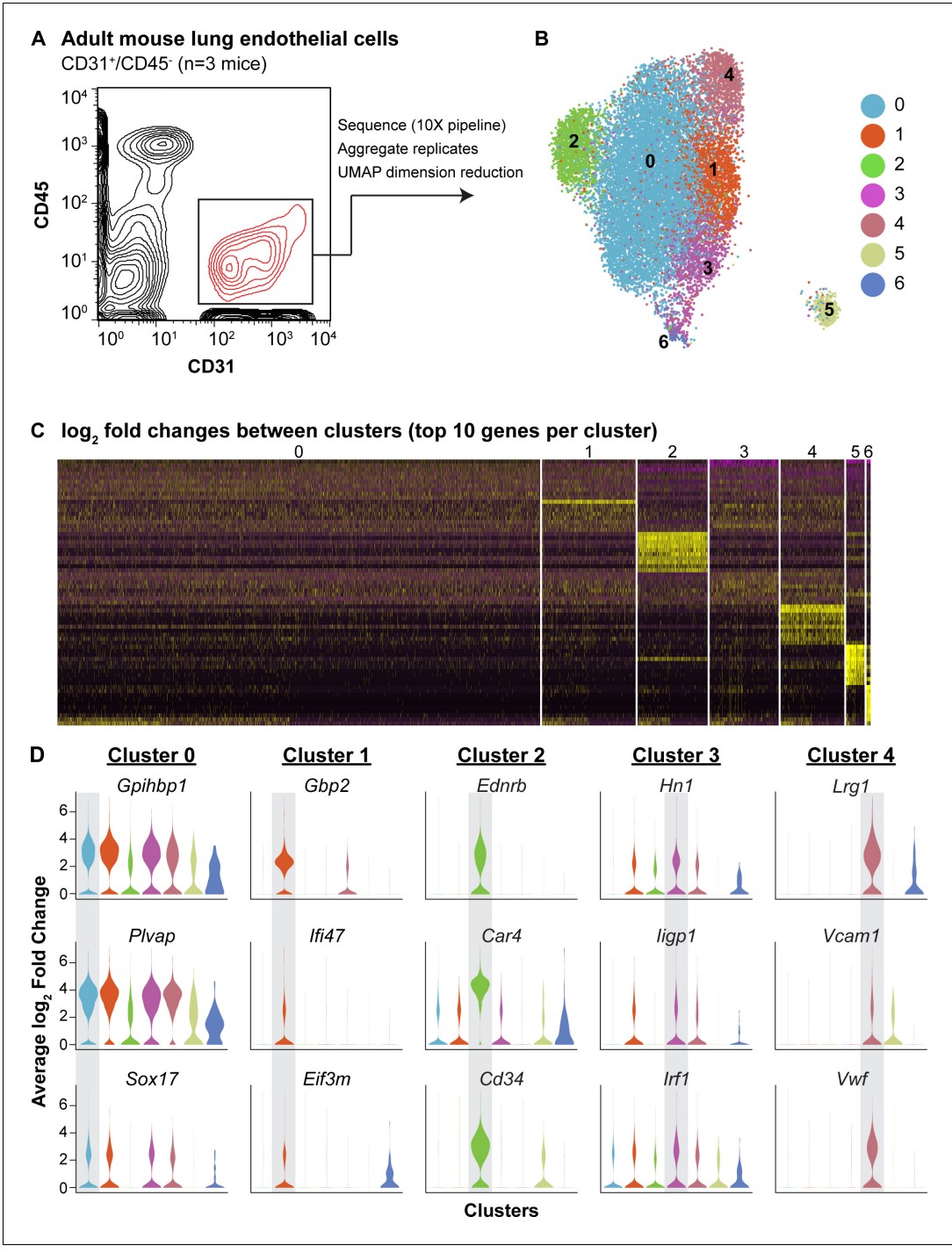

**Figure 1.** scRNA-seq analysis reveals heterogeneity of ECs in the adult mouse lung and identifies a distinct cell population high in *Car4* and *Cd34* expression. (**A**) Experimental strategy for identifying heterogeneity in adult mouse pulmonary endothelium. A representative FACS plot demonstrates gating used to isolate CD31$^+$/CD45$^-$ cells from adult mouse lung. (**B**) UMAP dimension reduction analysis of scRNA-seq data generated using a Seurat pipeline reveals seven distinct cell clusters within the CD31$^+$/CD45$^-$ cell population. (**C**) Analysis of the top 70 differentially expressed genes across all cells (top 10 differentially expressed genes per cluster by log$_2$ fold change in gene expression). (**D**) Violin plots show relative expression of representative, highly-expressed genes in each EC cluster across all clusters. Cluster two is defined by robust expression of *Ednrb*, *Car4*, and surface marker *Cd34*. See also *Figure 1—figure supplement 1*, *2*, *3*.

The online version of this article includes the following source data and figure supplement(s) for figure 1:

*Figure 1 continued on next page*

*Figure 1 continued*

**Figure supplement 1.** Isolation of CD45⁻/CD31⁺ cells results in capture of some mesenchymal and epithelial cells.

**Figure supplement 2.** Distinct contributions of individual mice to the integrated EC-specific scRNA-seq dataset.

**Figure supplement 3.** Isolation and qRT-PCR analysis of CD34-high ECs confirms their transcriptional distinction from other ECs.

**Figure supplement 3—source data 1.** Relative expression of endothelial cell genes in lung cell subpopulations isolated by FACS.

which are displayed using UMAP dimension reduction technique (*Figure 1B*; *Becht et al., 2018*). Each of the seven clusters was consistently identified across three biological replicates; clusters 5 and 6 represent a small amount of mesenchymal and epithelial contamination, respectively (*Figure 1—figure supplements 1* and *2*). Of the remaining 5 EC clusters, clusters 2 and 4 were the most distinct and exhibited significant transcriptional differences from the other clusters (*Figure 1C*). Cluster two was characterized by high expression of *Car4*, *Ednrb*, *Kdr*, and *Cd34* (*Figure 1D*; *Table 1*), whereas cluster four exhibited enhanced expression of genes related to maECs, such as *Vwf* and *Vcam1*. The other EC clusters (0, 1, and 3) expressed genes related to miECs, such as *Gpihbp1*, *Ifi47*, *Plvap*, and *Sox17* (*Figure 1D*). Isolation of cells comprising cluster two using a CD34 antibody (CD45⁻/CD31⁺/CD34-high) and comparison to other ECs (CD45⁻/CD31⁺/CD34-low) by qRT-PCR confirmed that at the population level, these cells express higher levels of *Cd34*, *Car4*, *Ednrb*, and *Kdr*, but comparable levels of other EC genes such as *Pecam1*, *Plvap*, *Gpihbp1*, and *Vwf* (*Figure 1—figure supplement 3*). The well-defined transcriptional signature of the cells composing cluster two indicates that the heterogeneity of the adult pulmonary endothelium at homeostasis extends beyond the distinction of maECs and miECs to include a transcriptionally distinct population of ECs with unknown functional relevance.

**Table 1.** Top 20 differentially expressed genes in *Car4*-high ECs compared to other ECs in scRNA-seq of FACS-isolated lung endothelium.

| Gene name | Average log fold change | Fraction of *Cd34*-high, *Car4*-high cells expressing this gene | Fraction of other ECs expressing this gene |
|---|---|---|---|
| Igfbp7 | 2.678286614 | 0.979 | 0.327 |
| Fibin | 2.574379173 | 0.776 | 0.058 |
| Car4 | 2.424088834 | 0.975 | 0.312 |
| Emp2 | 2.052891168 | 0.861 | 0.149 |
| Ednrb | 1.978598654 | 0.609 | 0.035 |
| AW112010 | 1.934007585 | 0.608 | 0.051 |
| Pmp22 | 1.684929983 | 0.889 | 0.299 |
| Ptp4a3 | 1.634843929 | 0.730 | 0.135 |
| Ccdc184 | 1.500061919 | 0.350 | 0.017 |
| Clu | 1.497660515 | 0.784 | 0.238 |
| Ccdc68 | 1.392943656 | 0.412 | 0.040 |
| Chst1 | 1.391322461 | 0.365 | 0.016 |
| Tmeff2 | 1.389485167 | 0.402 | 0.021 |
| Cd34 | 1.363682377 | 0.737 | 0.197 |
| Enho | 1.359701549 | 0.412 | 0.028 |
| Tbx2 | 1.300007606 | 0.376 | 0.027 |
| Rprml | 1.282823428 | 0.360 | 0.014 |
| Sept4 | 1.281675924 | 0.782 | 0.269 |
| Kdr | 1.233274427 | 0.571 | 0.123 |
| Apln | 1.225917934 | 0.360 | 0.022 |

## Gene Ontology and spatial restriction of endothelial subtypes

To understand differences in EC clusters at a functional level, we further characterized gene expression similarities and differences between clusters. A comparison of the overlap in Gene Ontology (GO) between clusters (*Figure 2A*) reveals similarity in both gene expression and GO between clusters 1 and 3 as well as between 3 and 4. Cluster two has little similarity with most other clusters, suggesting that this is a distinct cell population rather than a subset of a larger population. We also examined the top twenty GO categories across clusters (*Figure 2B*). Cluster one is uniquely enriched in genes involved in cellular response to interferon-beta, and cluster three is dominated by expression of genes that regulate translation, whereas cluster four exhibits high expression of genes involved in regulation of cell adhesion. Clusters 0 and 2 share enhanced expression of genes related to sprouting angiogenesis and blood vessel development. Cluster two is uniquely enriched in genes that are associated with vasculogenesis. Based on their gene expression signatures, we grouped the ECs into three main categories: 1) maECs (cluster 4), 2) miECs (clusters 0, 1 and 3), and *Car4*-high ECs (cluster 2) (*Figure 2C–E*). Although we would expect to isolate all ECs using a CD45$^-$/CD31$^+$ cell isolation strategy, we did not find *Prox1*$^+$ lymphatic ECs in this analysis, possibly because these cells were lost during single-cell isolation or preparation for sequencing.

To spatially locate the cell types of each cluster within the lung, we performed immunohistochemistry (IHC) for proteins encoded by representative genes expressed within each cluster. Immunostaining for Vwf and Vcam1 reveals that these proteins localize to the large vessel endothelium as previously reported (*Piali et al., 1995*; *Yamamoto et al., 1998*), with Vcam1 also present in the surrounding mesenchyme (*Figure 2C*). Immunostaining for Gpihbp1 and Plvap shows that both proteins are highly enriched in alveolar capillary plexus endothelium, with Plvap also present in large vessel endothelium, consistent with the scRNA-seq analysis (*Figure 2D*). IHC for CD34 and Car4 shows that both proteins are expressed in the alveolar region. The differences in gene expression between EC populations demonstrate that there may be functional relevance to heterogeneity within the pulmonary endothelium and within the alveolar capillary plexus.

## CD34-high endothelial population poised to contribute to regeneration

As the *Car4*-high EC population expressed high levels of *Vegfr2* and *Ednrb*, we sought to determine whether they contributed to alveolar regeneration following tissue damage. We employed an H1N1 influenza virus injury model, which causes heterogeneous injury similar to that of human influenza (*Kumar et al., 2011*; *Töpfer et al., 2014*; *Zacharias et al., 2018*). Control mice treated with intranasal PBS maintain normal alveolar structure (*Figure 3A,A'*), but the lungs of influenza-infected mice 14 days post infection (dpi) show regions of mild injury ('zones 1/2') (*Figure 3B,B'*) as well as regions of diffuse alveolar damage ('zone 3') (*Figure 3B,B''*) and total alveolar destruction ('zone 4') (*Figure 3B''*, dotted yellow line) (*Zacharias et al., 2018*). Evaluation of localization of CD31$^+$ and CD34$^+$ cells in control and damaged lungs by IHC revealed striking differences after influenza infection. In control lungs, CD34$^+$ cells were found distributed throughout the alveolar space, in a similar distribution to CD31$^+$/CD34-low miECs (*Figure 3C*). In alveolar space adjacent to the regions of diffuse alveolar damage after influenza infection, a dramatic increase in CD34 protein (*Figure 3D,E*) suggests that these cells may participate in repair and regeneration of damaged alveoli.

To determine whether the response of *Car4*-high ECs during regeneration occurs across different types of lung injury, we also employed an intratracheal bleomycin injury model (*Adamson, 1976*; *Adamson and Bowden, 1974*). Examination of bleomycin-injured lungs at 7, 14, and 21 days post injury (dpi) reveals extensive alveolar damage and inflammatory infiltration, with more extensive damage observed at 14 and 21 dpi (*Figure 3—figure supplement 1A–C,G*). Immunostaining for CD31 and CD34 demonstrated that CD34 expression in the areas of dense injury is located within compacted vessels and is more diffuse compared to control lungs (*Figure 3—figure supplement 1D–F*). Taken together, these data indicate that damage to the alveolar space through either influenza infection or bleomycin injury results in increased CD34$^+$ cells in areas adjacent to the greatest tissue damage, suggesting that these cells may play a role in regeneration of injured alveoli.

To further investigate the localization of EC subtypes at homeostasis and after influenza injury and to quantify numbers of *Car4*-high ECs across zones of injury, we employed RNAscope analysis of control and H1N1-injured tissue. Similar to IHC analysis for CD34/CD31, multiplex RNAscope analysis using probes against *Car4* and *Pecam1* revealed that *Car4*$^+$/*Pecam*$^+$ double-positive cells

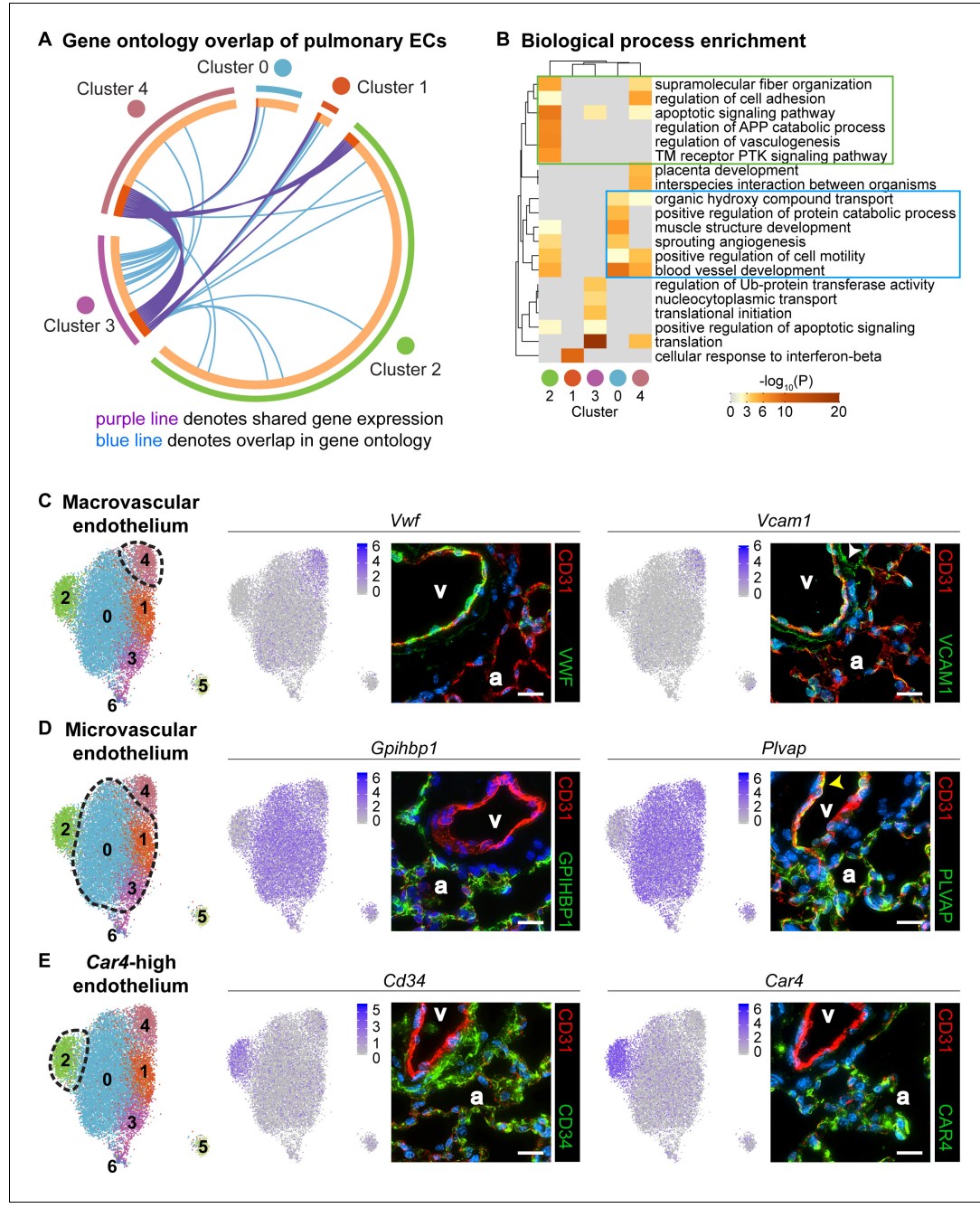

**Figure 2.** Gene ontology and localization of EC subtypes elucidates known functions and confirms divergence of clusters 2 and 4. (A) Circos plot shows gene expression and gene ontology (GO) overlap between distinct clusters of pulmonary ECs. A purple line connecting two clusters indicates expression of the same gene in both clusters, while a blue line connecting two clusters indicates expression of different genes found within the same GO category in each cluster. (B) GO biological process enrichment performed for each cluster and displayed in heatmap format demonstrates expression of genes associated with different biological processes in each cluster, including some overlap between clusters. Cluster 2 (green box) shares enrichment of some processes related to angiogenesis and blood vessel development with cluster 0 (blue box) but is distinct in its enrichment of genes related to vasculogenesis. (C) In-depth analysis of gene expression in cluster four indicates that this cluster likely represents macrovascular ECs (maECs) with high expression of *Vwf* and *Vcam1*. IHC indicates that these proteins localize mainly to the large vessel endothelium. White arrowhead demonstrates Vcam1 located in nearby mesenchymal cells. (D) Clusters 0, 1, and three represent a heterogeneous population of microvascular ECs (miECs) with high expression of *Gpihbp1* and *Plvap*. IHC indicates that these proteins localize to the alveolar capillary plexus endothelium. Yellow arrowhead demonstrates Plvap present in the large vessel endothelium. (E)

*Figure 2 continued on next page*

*Figure 2 continued*

Cluster two represents an as-yet uncharacterized population of ECs that localize to the alveolar region and express surface marker *Cd34* at a higher level from that in other ECs. These cells also express high levels of *Car4*. CD34 and Car4 proteins localize to the alveolar space, indicating a similar spatial distribution of cells in cluster two to that of miECs. v, vessel; a, alveolar space; scale bars in (C)-(E), 20 microns.

localize throughout the alveolar space at homeostasis; however, not all *Pecam1*⁺ cells in the alveolar space are *Car4*⁺ (*Figure 3—figure supplement 2A*). A subset of both *Car4*-high (*Car4*⁺/*Pecam*⁺ double-positive) ECs and other (*Car4*⁻/*Pecam*⁺) ECs are found adjacent to *Hopx*⁺ AT1 cells as well as *Sftpc*⁺ alveolar type 2 (AT2) epithelial cells (*Figure 3—figure supplement 3*). This analysis does not suggest, but also does not rule out, a preferential interaction of *Car4*-high ECs with AT1 or AT2 cells. At 14 days post H1N1 influenza injury, quantitative analysis of *Car4*⁺/*Pecam*⁺ double-positive cells in different zones of injury (*Zacharias et al., 2018*) demonstrated an increase in number of double-positive cells in regions of mild to moderate injury (zones 1/2) (*Figure 3—figure supplement 2B,D*) as well as in regions adjacent to severe alveolar damage (zone 3) (*Figure 3—figure supplement 2C,D*). Relatively few double-positive cells were observed in regions of alveolar destruction (zone 4) (*Figure 3—figure supplement 2D*), and we chose to focus on zones 1–3, as Krt5⁺ cell expansion after influenza infection has been reported to result in hypoxic vasoconstriction in regions of severe alveolar damage (*Xi et al., 2017*). Prior to injury, *Gpihbp1*⁺ miECs are also localized throughout the alveolar space. However, *Gpihbp1*⁺/*Plvap*⁺/*Pecam1*⁺ triple-positive cells were not found in the large vessel endothelium, where cells expressed only *Plvap* and *Pecam1* (*Figure 3—figure supplement 4A*). Further analysis of macrovascular ECs using probes against *Cxcl12*, expressed by arterial maECs, or *Vegfc*, expressed by venous maECs, reveals that these cells localize to large vessels both in control animals (*Figure 3—figure supplement 4B,C*) and at 14 dpi (*Figure 3—figure supplement 4E,F*). *Gpihbp1*⁺/*Plvap*⁺/*Pecam1*⁺ triple-positive miECs remain localized to the alveolar space at 14 dpi, with a greater density of these cells in injured tissue regions (*Figure 3—figure supplement 4D*). These data suggest that both *Car4*-high ECs and *Gpihbp1*⁺ miECs may play a role in response to alveolar injury based on changes to their localization or density during the regenerative period following H1N1 influenza infection.

## Changes to endothelial cell populations and transcriptomes following influenza infection

To understand how *Car4*-high ECs and other EC populations behave following influenza injury in the context of the entire lung and whether they undergo gene expression changes that may contribute to regeneration, we performed scRNA-seq on non-immune cells (CD45⁻) isolated from whole adult mouse lungs, both uninjured and at 14 days following H1N1 influenza infection. Histological examination of the lungs at 14 dpi revealed regions of damage compared to control lung and an increase in CD45-expressing cells indicative of an immune response to the influenza injury (*Figure 4A,B*). We identified endothelial, epithelial, and mesenchymal cell clusters in the adult mouse lung at homeostasis and at 14 dpi, displayed using UMAP dimension reduction (*Figure 4C,D*; *Figure 4—figure supplements 1* and *2*; *Becht et al., 2018*). Comparison of expression of representative genes for each cluster across all clusters demonstrates that these clusters are indeed distinct cell populations (*Figure 4—figure supplement 3*). Analysis of the most highly-expressed genes in each endothelial cluster demonstrated that *Car4*-high ECs, two populations of miECs (*Gpihbp1*-high and *Ifi47*-high), and both venous and arterial maECs could be identified in both uninjured and injured lung (*Figure 4E,F*). Of note, the whole-lung scRNA-seq analysis identified additional EC populations in comparison to the CD31⁺ sorted cells, likely due to a less biased isolation procedure as well as including the presence of additional non-ECs, which provides greater differences in number and transcriptional variance of cell types analyzed. For instance, the whole-lung scRNA-seq analysis generated separate arterial and venous maEC clusters. Moreover, lymphatic ECs were identified in both control and H1N1 whole-lung datasets. scRNA-seq analysis at 14 dpi also revealed a newly emergent cluster not observed at homeostasis, a population of proliferative ECs (*Figure 4D,F*). Proliferating ECs expressed high levels of markers of proliferation such as *Mki67*, *Ccnb2*, *Cdk1*, and *Cdc20* (*Table 2*), but also expressed other EC markers such as *Cd31*, *Gpihbp1*, *Plvap*, and *Cd34* (*Figure 4—figure supplement 4*). Interestingly, proliferating ECs did not express high levels of genes expressed

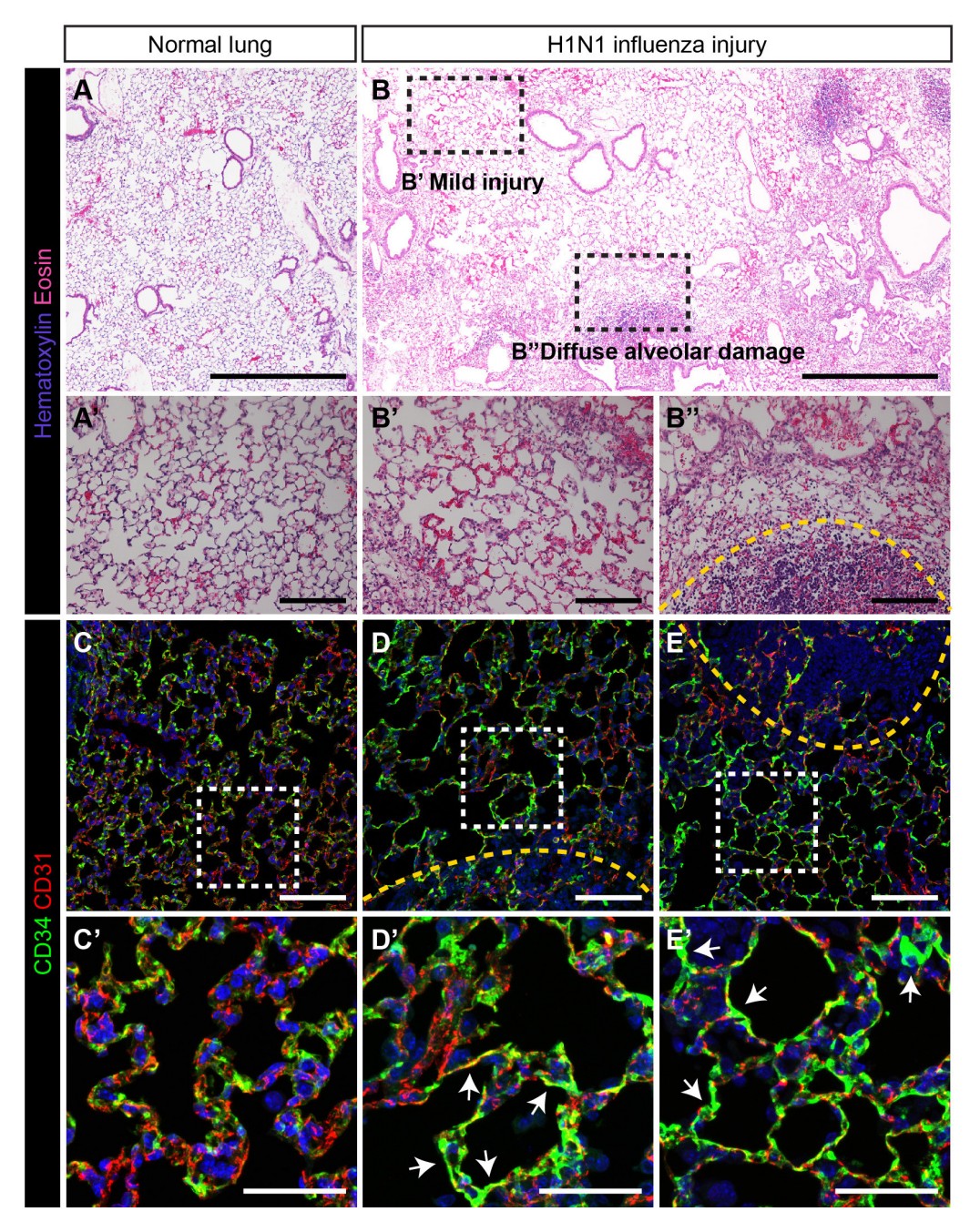

**Figure 3.** CD34-expressing ECs respond robustly to H1N1 influenza injury. In comparison to (**A**) normal adult mouse lung, (**B**) influenza-injured lungs demonstrate heterogeneous, regional alveolar damage, including regions of mild (zone 1/2) (**B'**) and severe (zone 3) (**B''**) injury (*Zacharias et al., 2018*). A region of total alveolar destruction (zone 4) is outlined with a yellow dotted line. Scale bars in (**A**), (**B**), 500 microns. Scale bars in (**A'**), (**B'**), and (**B''**), 100 microns. (**C, C'**) In control lung, CD34-expressing cells are located in the alveolar space, with a similar distribution to that of CD31-expressing miECs. (**D–E**) However, in zone three regions adjacent to areas of severe alveolar damage indicated by yellow dotted lines, a dramatic increase in CD34 localization by IHC suggests that these cells are recruited to sites of alveolar injury. Images are representative of N = 4 animals (H1N1 influenza injury) and N = 3 animals (control). Scale bars in (**C**)-(**E**), 100 microns. Scale bars in (**C'**)-(**E'**), 50 microns.
The online version of this article includes the following source data and figure supplement(s) for figure 3:

**Figure supplement 1.** Response of CD34-expressing cells to bleomycin injury is comparable to their response to influenza injury.

*Figure 3 continued on next page*

*Figure 3 continued*

**Figure supplement 2.** *Car4*-expressing ECs are increased in regions of mild to severe injury during the response to H1N1 influenza infection in the lung.

**Figure supplement 2—source data 1.** Cell counts for total number of Car4-expressing, Pecam1-expressing cells, normalized to total number of nuclei.

**Figure supplement 3.** Localization of *Car4*-high ECs and other ECs with alveolar type 1 and type two epithelial cells.

**Figure supplement 3—source data 1.** Percentage of *Car4*-high EC within 3 microns of Hopx-expressing AT1 cells or Sftpc-expressing AT2 cells.

**Figure supplement 4.** Pulmonary miECs and maECs do not change localization during response to H1N1 influenza infection.

by *Car4*-high ECs, such as *Kdr* and *Ednrb*, or of genes expressed by maECs, such as *Vcam1*, or *Vwf* (*Figure 4—figure supplement 4*).

To determine the changes in gene expression that occur in heterogeneous populations of ECs with H1N1 injury, we aggregated individual control and flu scRNA-seq datasets to generate an integrated dataset (*Figure 4—figure supplement 5A–B*). Each individual control and H1N1-injured animal contributed to each cluster identified in the integrated dataset, with the exception of the proliferating EC population, which was composed mainly of cells from the two H1N1 datasets (*Figure 4—figure supplement 5B–E*). Gene expression analysis of known marker genes, or genes used to identify epithelial, mesenchymal, and endothelial cell types in individual scRNA-seq datasets, also allowed us to identify these clusters in the integrated dataset (*Figure 4—figure supplement 5F*), confirming a direct correlation between cell types identified in the uninjured and injured lungs.

We further probed the integrated scRNA-seq dataset to determine how the transcriptome of EC populations may change after H1N1 injury. We first focused on *Car4*-high ECs. The gene expression profile of this cell type within the influenza dataset (*Table 3*) appeared similar to that observed at homeostasis (*Table 1*), suggesting that these cells do not differ appreciably in gene expression following influenza. However, analysis of 12 of the top differentially expressed genes in this cluster between control and flu datasets revealed small but statistically significant decreases in expression in signaling proteins such as *Acvrl1*, *Adgre5*, and *Thbd* (thrombomodulin, an endothelial-specific receptor that binds to thrombin), as well as in the hypoxia-inducible transcription factor *Epas1* and in *Emp2*, which positively regulates *Vegfa*. Genes with statistically significant increases in transcription after H1N1 influenza infection in this cluster included genes that regulate the cytoskeleton and extracellular matrix, such as *Sept4* and *Sparc*, as well as the transcription factor *Hes1* (*Figure 4—figure supplement 6A*). These expression differences suggest that in addition to an increase in number of *Car4*-high ECs in H1N1-injured regions of the distal lung, transcriptional changes within this cell population may contribute to regeneration. However, as expression of the genes that define this cell population in comparison to other cell types (*Tables 1* and *3*) does not change significantly after injury, it is likely that changes in cell fate are not a dominant component of the injury response of this population. To determine global changes in gene expression across EC populations after influenza infection, we performed GO analysis on all genes differentially expressed between H1N1 and control scRNA-seq datasets across EC populations (*Figure 4—figure supplement 6B*). This analysis revealed that ECs respond to flu by increasing transcription of genes related to translation, signaling to promote survival and proliferation, cellular response to stress, and cell death.

## Endothelial cell proliferation increases significantly during H1N1 injury response

The emergence of proliferating ECs following influenza injury led us to probe proliferation throughout the lung during at this stage of the regenerative process. We investigated the extent of EC proliferation after influenza injury by EdU incorporation in ECs from both control and H1N1-injured mice. After intranasal administration of H1N1 or PBS (control), we gave mice EdU in their drinking water either from days 0–7 (H1N1 cohort 1), days 7–14 (H1N1 cohort two and control cohort), or days 14–21 (H1N1 cohort 3) (*Figure 5A*). To ensure that there were no cytotoxic effects of EdU when administered over the 7 day time course, we compared proliferation by IHC for Ki67 or phospho-histone H3 in EdU-treated and -untreated animals at 14 dpi and observed no difference

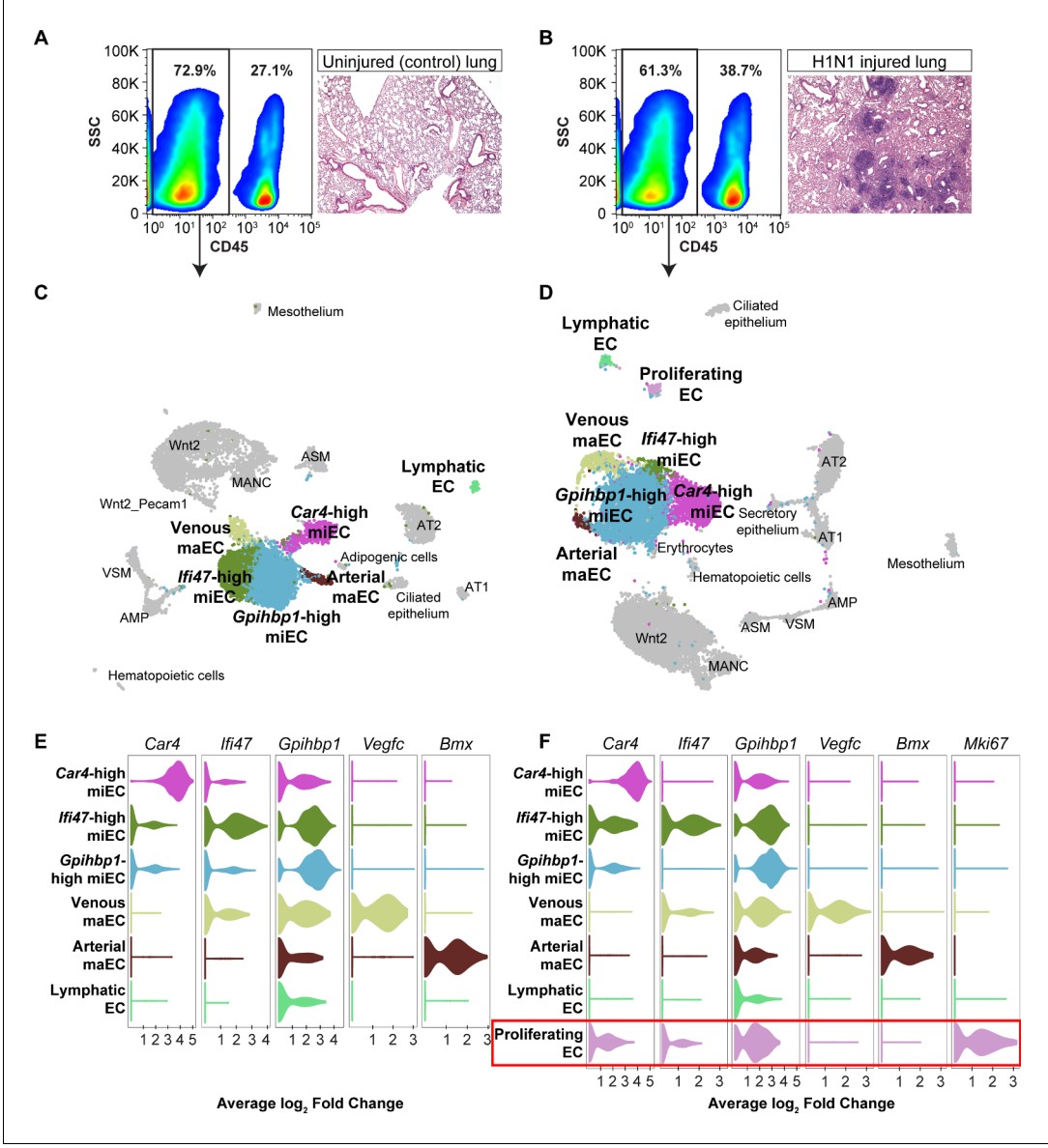

**Figure 4.** scRNA-seq analysis of whole lungs reveals the appearance of a proliferating EC population in the adult mouse lung after influenza injury. Representative FACS plots demonstrate gating used to isolate CD45⁻ cells from (**A**) control and (**B**) influenza-infected adult mouse lungs and indicate an increase in CD45⁺ cells after influenza infection. H and E staining demonstrates regions of dense injury in H1N1-injured lung, visible as dark purple regions in the alveolar space. Clustering and UMAP dimension reduction of scRNA-seq data from (**C**) control and (**D**) H1N1-injured lung identifies epithelial, mesenchymal, and EC clusters. EC populations in both datasets include venous and arterial maECs, two populations of miECs, lymphatic ECs, and *Car4*-high ECs. In addition, a population of proliferating ECs that is not present in the control mouse lung arises after influenza injury. Violin plots show expression of representative genes for miEC, maEC, and *Car4*-high EC clusters across ECs in (**E**) control and (**F**) H1N1-injured lung datasets. Red box indicates the newly emergent proliferating EC population. The online version of this article includes the following figure supplement(s) for figure 4:

**Figure supplement 1.** Epithelial cell clusters in control and influenza-infected adult mouse lung.
**Figure supplement 2.** Mesenchymal cell clusters in control and influenza-infected adult mouse lung.
**Figure supplement 3.** Gene expression of representative genes for each cluster across all clusters.
**Figure supplement 4.** Proliferating ECs express general EC marker genes, but do not express genes that are highly expressed in *Car4*-high ECs.
**Figure supplement 5.** An integrated scRNA-seq dataset reveals concordance between cell types identified in individual control and H1N1 injury datasets.

*Figure 4 continued on next page*

*Figure 4 continued*

**Figure supplement 6.** Analysis of transcriptional changes within EC clusters after H1N1 influenza injury.

(*Figure 5—figure supplement 1*). At the endpoint of EdU administration for each cohort, we isolated single cells from the mouse lungs and quantified %CD45$^+$ immune cells, %CD45$^-$/CD31$^+$ ECs, and %EdU$^+$ ECs using flow cytometry. As expected, H1N1 injury results in a dramatic increase in immune cells at all time points (*Figure 5B,H*). However, the total number of CD31$^+$ ECs did not significantly differ between control and flu cohorts (*Figure 5C,I*), possibly because injury results in decreased ECs at a time point earlier than seven dpi. Between 7 and 14 dpi, a large proportion of ECs from H1N1-treated mice had incorporated EdU (CD45$^+$/CD31$^+$/EdU$^+$ cells) (*Figure 5B–D*) compared to PBS treated mice (*Figure 5—figure supplement 2A–C*). To determine which EC populations contribute to this proliferative response, we further separated CD45$^-$/CD31$^+$ ECs into CD34-high and CD34-low populations (*Figure 5E*), representing *Car4*-high ECs and other ECs, respectively. H1N1-injured mice demonstrated EdU incorporation in both EC subpopulations from 7 to 14 dpi (*Figure 5F,G*), whereas control animals showed very little EdU incorporation in either population (*Figure 5—figure supplement 2D–F*). By comparing these results to those of injured mice administered EdU from days 0–7 of response to H1N1 infection (*Figure 5—figure supplement 3*) or from days 14–21 of response to H1N1 infection (*Figure 5—figure supplement 4*), we found that EC proliferation in general (*Figure 5J*), as well as proliferation within specific CD34-high and CD34-low EC populations (*Figure 5L,M*) increases significantly over the course of the injury response. Further, we observed a significant increase in the proportion of CD34-high ECs over the course of the regenerative response (*Figure 5K*), which aligns with our observation that these cells increase in regions of mild to severe damage in the lung tissue after injury. Taken together, these data indicate that both

**Table 2.** Top 20 differentially expressed genes in proliferating ECs compared to other cell populations in scRNA-seq of whole mouse lung after influenza infection.

| Gene name | Average log fold change | Fraction of proliferating ECs expressing this gene | Fraction of all cells expressing this gene |
|---|---|---|---|
| *Ube2c* | 2.317853367 | 0.839 | 0.011 |
| *Top2a* | 2.253705903 | 0.878 | 0.011 |
| *Prc1* | 2.162682748 | 0.863 | 0.012 |
| *Rrm2* | 2.137769719 | 0.671 | 0.016 |
| *Cenpf* | 2.120348153 | 0.863 | 0.015 |
| *Cks2* | 2.119347813 | 0.886 | 0.064 |
| *Pbk* | 2.101722647 | 0.788 | 0.008 |
| *Cdc20* | 2.055640843 | 0.769 | 0.017 |
| *Nusap1* | 2.04083334 | 0.898 | 0.008 |
| *Birc5* | 1.983612047 | 0.847 | 0.009 |
| *Cdk1* | 1.951263566 | 0.808 | 0.014 |
| *Smc4* | 1.830329324 | 0.878 | 0.124 |
| *Spc25* | 1.79466608 | 0.769 | 0.012 |
| *Cenpa* | 1.78639737 | 0.753 | 0.027 |
| *Smc2* | 1.74071554 | 0.808 | 0.051 |
| *Ccnb2* | 1.7002509 | 0.745 | 0.013 |
| *Lockd* | 1.699667491 | 0.784 | 0.024 |
| *Lmnb1* | 1.640231709 | 0.792 | 0.04 |
| *Fam64a* | 1.620370016 | 0.741 | 0.005 |
| *Racgap1* | 1.579862823 | 0.725 | 0.009 |

**Table 3.** Top 20 differentially expressed genes in *Car4*-high ECs compared to other cell populations in scRNA-seq of whole mouse lung following influenza infection.

| Gene name | Average log fold change | Fraction of *Car4*-high ECs expressing this gene | Fraction of all cells expressing this gene |
|---|---|---|---|
| *Car4* | 2.647460241 | 0.981 | 0.281 |
| *Fibin* | 2.226759092 | 0.83 | 0.133 |
| *Ednrb* | 2.153211962 | 0.834 | 0.057 |
| *AW112010* | 2.041819815 | 0.934 | 0.259 |
| *Cyp4b1* | 1.822454969 | 0.986 | 0.515 |
| *Emp2* | 1.774097564 | 0.957 | 0.439 |
| *Ptp4a3* | 1.704881479 | 0.83 | 0.161 |
| *Kitl* | 1.522369803 | 0.87 | 0.258 |
| *Ly6c1* | 1.517128641 | 0.993 | 0.546 |
| *Kdr* | 1.514651153 | 0.835 | 0.194 |
| *Cd34* | 1.465370921 | 0.92 | 0.372 |
| *Apln* | 1.450197164 | 0.567 | 0.032 |
| *Icam2* | 1.438572854 | 0.981 | 0.478 |
| *Tspan13* | 1.437697131 | 0.993 | 0.585 |
| *Ly6a* | 1.420070001 | 0.997 | 0.647 |
| *Clu* | 1.398326741 | 0.905 | 0.343 |
| *Ecscr* | 1.346567228 | 0.953 | 0.446 |
| *Nrp1* | 1.310196884 | 0.974 | 0.639 |
| *Chst1* | 1.303951311 | 0.517 | 0.028 |
| *Ccdc184* | 1.272517358 | 0.44 | 0.021 |

*Car4*-high ECs and other ECs contribute to the proliferative response of the endothelium after acute lung injury.

To determine the localization of proliferating ECs within the lung, we performed RNAscope analysis both at homeostasis and after H1N1 injury. In the control lung, *Pecam1*-expressing ECs coexpressing proliferation genes *Top2a* and *Prc1* are extremely rare (**Figure 5—figure supplement 5A, D**). However, at 14 days post-H1N1 injury, *Top2a*[+]/*Pecam1*[+]/*Prc1*[+] triple-positive cells are significantly increased within regions of mild to moderate alveolar injury (zones 1/2) (**Figure 5—figure supplement 5B,D**) as well as within regions of severe injury (zone 3) (**Figure 5—figure supplement 5C, D**) adjacent to areas of total alveolar destruction. This indicates that endothelial proliferation likely contributes to repair of the pulmonary vasculature across the injured tissue.

We also performed additional analysis of our scRNA-seq datasets to determine predicted cell numbers in G1, G2/M, and S phases for each cluster based on expression of genes associated with each cell cycle phase at homeostasis and after influenza injury. Of note, this is a predictive analysis rather than a functional one and does not confirm that cells are actively cycling. In the uninjured lung, cells are predicted to be evenly distributed throughout these three states (**Figure 5—figure supplement 6A,C**). At 14 dpi, however, a larger percentage of ECs compared to epithelial, mesenchymal, and other cell types are predicted to be in G2/M and S phases based on their gene expression profiles (**Figure 5—figure supplement 6B,D**). Examination of individual endothelial cell types at homeostasis and after influenza injury demonstrates that the predicted cell cycle profiles of miECs, maECs, and *Car4*-high ECs are similar to each other at homeostasis as well as at 14 dpi but reveals an extremely high number of cells predicted to be in G2/M phase in the newly emergent proliferating EC cluster (**Figure 5—figure supplement 6E,F**). In combination with functional proliferation studies, these transcriptomic data indicate that no individual cluster of ECs possesses preferential proliferative potential and demonstrate the ability of scRNA-seq data to predict the presence of a highly proliferative cell population.

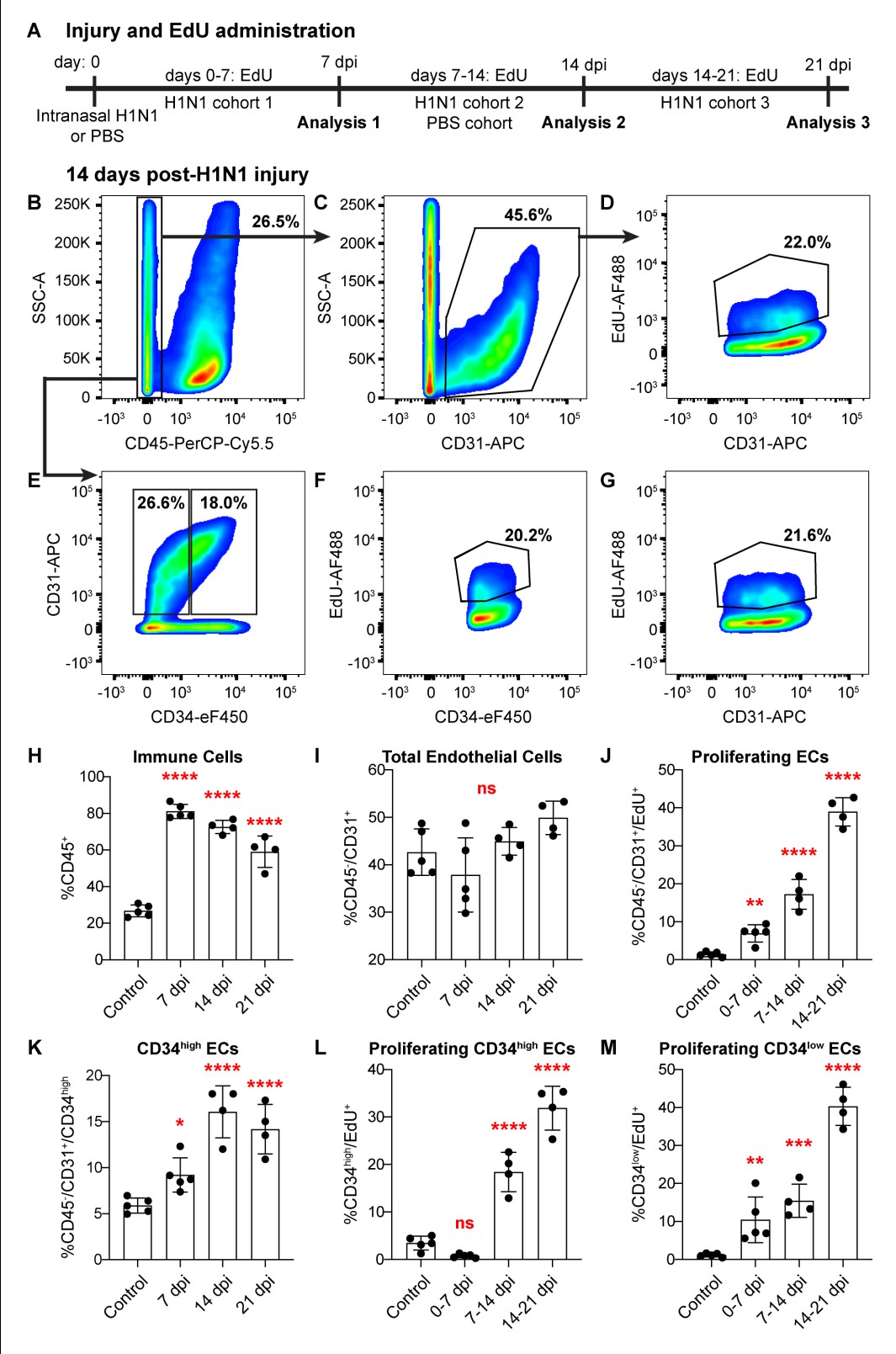

**Figure 5.** Proliferation of lung ECs increases dramatically during the regenerative response to H1N1 influenza infection. (**A**) Outline of the experimental strategy used to assess EC proliferation. Mice given intranasal H1N1 or PBS (control) were given EdU ad libitum in their drinking water (0.2 g/L) either from days 0–7 following infection (N = 5), days 7–14 following infection (N = 4) or PBS (N = 5), or days 14–21 following infection (N = 4). On the last

*Figure 5 continued on next page*

*Figure 5 continued*

day of EdU administration, mice were euthanized and single-cell suspensions were generated for flow cytometry analysis. (B) At 14 days post-H1N1 injury, single cells isolated from the lungs of a representative mouse given EdU from days 7–14 were 26.5% CD45-negative non-immune cells, (C) of which 45.6% were CD31-positive ECs. (D) Of these ECs, 22.0% had incorporated EdU over days 7–14 of the regenerative response to H1N1 injury. (E) Separation of CD45-negative, CD31-positive ECs into CD34-high and CD34-low populations reveals that in the same representative mouse, 18% of non-immune cells were CD34-high ECs. Within these EC subpopulations, (F) 20.2% of CD34-high ECs and (G) 21.6% of CD34-low ECs had incorporated EdU over days 7–14 of the regenerative response. (H) Quantification of %CD45-positive immune cells over injury and control cohorts demonstrates a significant increase in immune cell populations within the lung at 7, 14, and 21 days post-influenza injury. (I) Quantification of %CD45-negative/CD31-positive ECs in injury and control cohorts shows no significant difference in EC numbers at 7, 14, and 21 days post-influenza injury, perhaps because H1N1 injury depletes ECs at an earlier time point after injury. (J) Quantification of %CD45-negative/CD31-positive/EdU-positive proliferating ECs across all three time windows demonstrates a significant increase in EC proliferation between control animals administered EdU for 7 days and all three time windows considered after H1N1 injury. (K) The percentage of % CD45-negative/CD31-positive/CD34-high ECs also increases significantly at days 7, 14, and 21 following H1N1 injury. When EdU incorporation is considered individually in (L) CD34-high ECs and (M) CD34-low ECs, both populations proliferate significantly during the H1N1 injury response. *, $p < 0.05$; **, $p < 0.01$; ***, $p < 0.001$; ****, $p < 0.0001$; ns, not significant.

The online version of this article includes the following source data and figure supplement(s) for figure 5:

**Figure supplement 1.** Assessment of proliferation by IHC for phospho-histone H3 reveals no differences between EdU-treated and EdU-untreated mice.

**Figure supplement 1—source data 1.** Cell counts for total number of proliferating cells, normalized to total number of nuclei.

**Figure supplement 2.** ECs proliferate very little in uninjured mice.

**Figure supplement 3.** Proliferation in CD34-low ECs increases in the first seven days following H1N1 injury.

**Figure supplement 4.** Endothelial proliferation continues to increase from 14 to 21 days following H1N1 injury.

**Figure supplement 5.** Proliferating ECs are located across regions of mild to severe injury after H1N1 influenza infection.

**Figure supplement 5—source data 1.** Cell counts for total number of Top2a- and Prc1-expressing endothelial cells and non-endothelial cells, normalized to total number of nuclei.

**Figure supplement 6.** Expression of cell cycle genes in scRNA-seq transcriptomes predicts high levels of proliferation in pulmonary ECs after influenza injury.

**Figure supplement 6—source data 1.** Percentage of cells predicted to be in G1, G2/M, or S phases based on their gene expression profiles.

---

To further elucidate the origins of proliferating ECs after influenza injury, we used pseudotime analysis of endothelial populations within our H1N1 scRNA-seq dataset. The Slingshot program (*Street et al., 2018*) was used to infer trajectories within EC populations after influenza infection and demonstrated a close association between miECs and proliferating ECs (*Figure 6A,B*), in agreement with our data demonstrating low expression of *Cd34* and other markers of *Car4*-high ECs as well as high expression of miEC markers in proliferating ECs (*Figure 4—figure supplement 4*). Two different lineage trajectories were revealed, with curve one connecting *Car4*-high ECs, miECs, and proliferating ECs and curve two connecting *Car4*-high ECs to a different subset of miECs (*Figure 6C–D*). Gene expression analysis along each trajectory indicates that the top genes defining expression differences along curve one include genes highly expressed in *Car4*-high ECs, such as *Ednrb*, *Tbx2*, *Cd34*, *Car4*, and *Kdr*, as well as cell cycle genes highly expressed in proliferating ECs, such as *Ube2c*, *Cdk1*, and *Top2a* (*Figure 6E*; *Figure 6—figure supplement 1*). The top genes defining expression differences along curve two also include genes highly expressed in *Car4*-high ECs, as well as genes expressed more generally throughout miECs, such as *Atf3* and *Nrp1* (*Figure 6F*; *Figure 6—figure supplement 2*). In combination, EdU incorporation analysis and pseudotime analysis suggest that although both CD34-high and CD34-low ECs proliferate after H1N1 injury, lineage relationships between EC subpopulations during the regenerative process may not be straightforward. These data allow for the possibility that CD34-high, *Car4*-high cells must undergo transcriptional changes in order to proliferate, or that proliferating miECs upregulate CD34 as they move along a differentiation trajectory to generate *Car4*-high ECs. *Car4*-high ECs may therefore represent a more differentiated or specialized cell type that participates in the regenerative process by signaling to

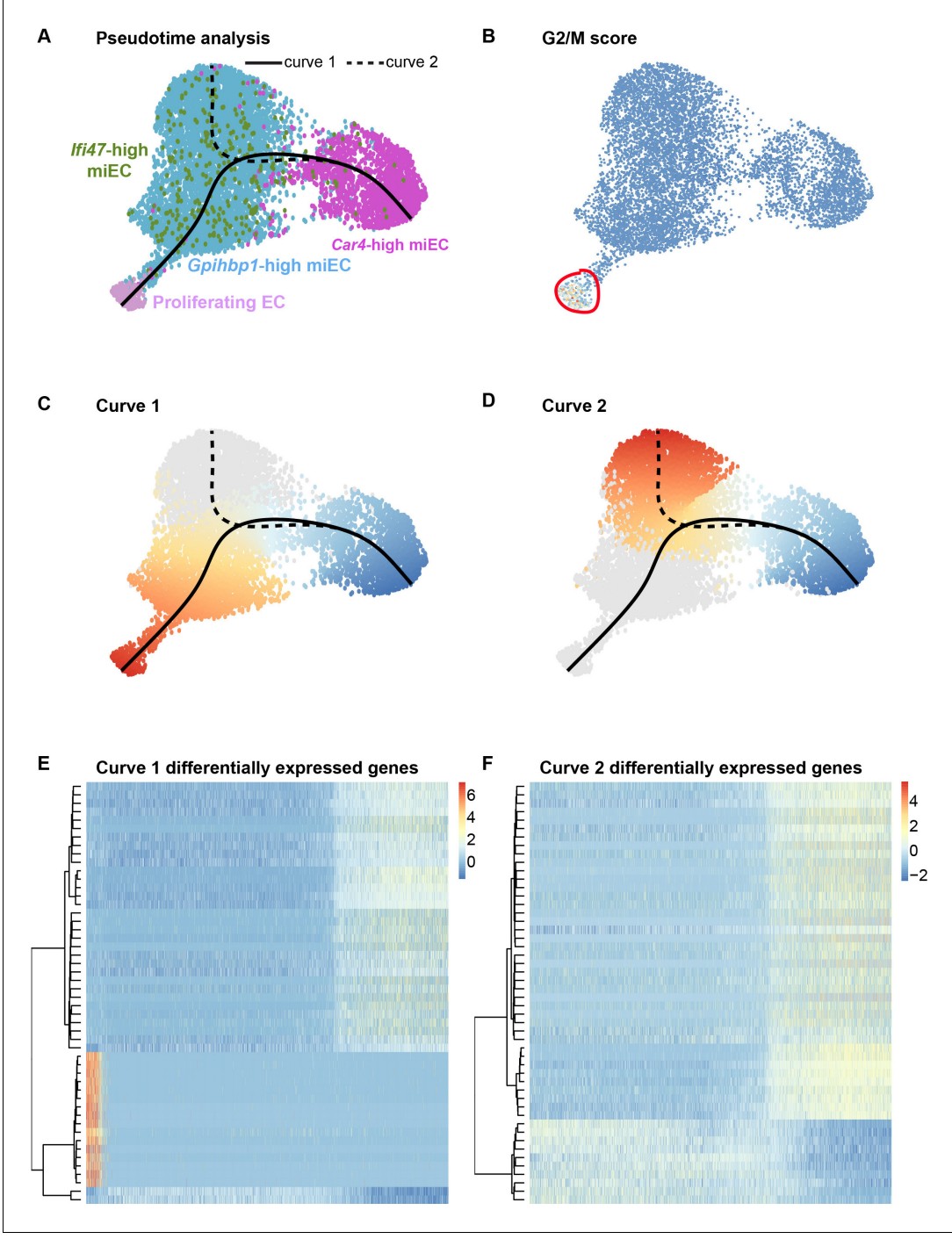

**Figure 6.** Proliferating ECs are more closely related to miECs than to *Car4*-high ECs. (**A**) Pseudotime analysis of *Car4*-high EC, *Gpihbp1*-high miEC, *Ifi47*-high miEC, and proliferating EC clusters at 14 dpi using Slingshot identifies putative lineage relationships between the clusters based on gene expression similarities. Proliferating ECs are more similar in gene expression to both miEC populations than to *Car4*-high ECs. (**B**) Proliferating ECs have the highest G2/M score among all four populations, with orange/red indicating cells predicted to be in G2/M and blue indicating cells not predicted to be in G2/M. (**C**), (**D**) Slingshot identifies two lineage trajectories among the four EC populations. Curve one connects *Car4*-high ECs and proliferating ECs through a miEC intermediate, while curve two connects *Car4*-high ECs and miECs. Color indicates direction of the trajectory, which is defined by the program and proceeds from blue to red. (**E**), (**F**) Heatmaps displaying the top 50 differentially expressed genes along curves 1 and 2. Curve one demonstrates the greatest variance in gene expression along the trajectory,

*Figure 6 continued on next page*

*Figure 6 continued*

indicating greater gene expression differences between *Car4*-high ECs and proliferating ECs than between *Car4*-high ECs and miECs.

The online version of this article includes the following figure supplement(s) for figure 6:

**Figure supplement 1.** Feature plots demonstrate the expression level of each of the top 15 differentially expressed genes along curve 1 of the Slingshot pseudotime trajectory, superimposed on UMAP space to demonstrate cluster(s) in which each gene is expressed.

**Figure supplement 2.** Feature plots demonstrate the expression level of each of the top 15 differentially expressed genes along curve 2 of the Slingshot pseudotime trajectory, superimposed on UMAP space to demonstrate cluster(s) in which each gene is expressed.

---

other endothelial subsets or to epithelial and mesenchymal lineages to coordinate cell behaviors in the alveolar niche after injury, whereas endothelial proliferation is a capability possessed by all ECs.

## Ligand-receptor analysis indicates crosstalk between *Car4*-high ECs and the alveolar epithelium

To further interrogate how the signaling abilities of *Car4*-high ECs may contribute to alveolar regeneration, we performed ligand-receptor interaction analysis based on our scRNA-seq dataset. We compared possible interactions of *Car4*-high ECs, proliferating ECs, and miECs, both with each other and with the primary epithelial lineages in the alveolus. We found that at homeostasis, the highest number of putative epithelial-endothelial interactions are predicted to occur between ligand-producing AT1 epithelial cells and *Car4*-high ECs or *Gpihbp1*-high miECs (*Figure 7A,C*). After H1N1 influenza injury, our data indicate that signaling interactions between AT1 cells and *Car4*-high ECs or miECs are predicted to remain high (*Figure 7B*). The emergent proliferating EC population is also predicted to participate in signaling interactions with AT1 cells as well as with miECs and *Car4*-high ECs after H1N1 influenza infection (*Figure 7D*). These predictions indicate that endothelial-epithelial interactions are likely not restricted to *Car4*-high ECs, but rather are a general property of the capillary endothelium during the response to injury.

We examined putative AT1-*Car4*-high EC interaction mediators in more detail and found that these included ECM interactions, such as between collagen and integrin alpha and beta subunits or between metalloproteases and integrins. We also identified predicted semaphorin-neuropilin 1 (Nrp1) interactions, Bmp4-Bmpr2 interactions, and Fgf1-Fgfr3 interactions in addition to Vegfa-Vegfr1 (Flt1), Vegfa-Vegfr2 (Kdr), and Vegfa-neuropilin 1 (Nrp1) interactions (*Table 4*). Importantly, many of these signaling pathways have been identified as essential for vascular development and angiogenesis, and some are misregulated in pulmonary disease (*Asahara et al., 1999b*; *Blanco and Gerhardt, 2013*; *Chao et al., 2019*; *Lee et al., 2014*; *Londhe et al., 2011*; *Nguyen et al., 2005*; *Potente and Mäkinen, 2017*; *Rojas-Quintero et al., 2018*; *Simons et al., 2016*; *White et al., 2007*). Future studies to validate signaling interactions between ECs and other pulmonary cell types will allow for a more conclusive statement on the importance of this and other signaling axes in lung regeneration. In combination, our scRNA-seq data and in vivo functional validation have identified a unique population of *Car4*-high ECs that contribute to endothelial regeneration in the lung through proliferation and predicted signaling interactions with AT1 cells, which may be essential in both regulating homeostasis and repair of the gas-exchange interface.

## Discussion

Through single-cell analysis of the pulmonary vasculature at homeostasis and after acute lung injury, we have expanded our understanding of vascular heterogeneity in the lung and begun to elucidate the cellular mechanisms by which the vasculature contributes to alveolar regeneration after injury. In combination with transcript and protein localization, our scRNA-seq analysis identified a unique population of microvascular endothelium, the *Car4*-high EC. Through further characterization of these cells, we have determined that they coalesce within sites of moderate-to-severe alveolar damage after influenza infection and possess a transcriptional signature suggesting they are primed to receive signals from other alveolar cell types. Gene ontology analysis also suggests a role for these cells in vasculogenesis and blood vessel development. We additionally demonstrate the emergence

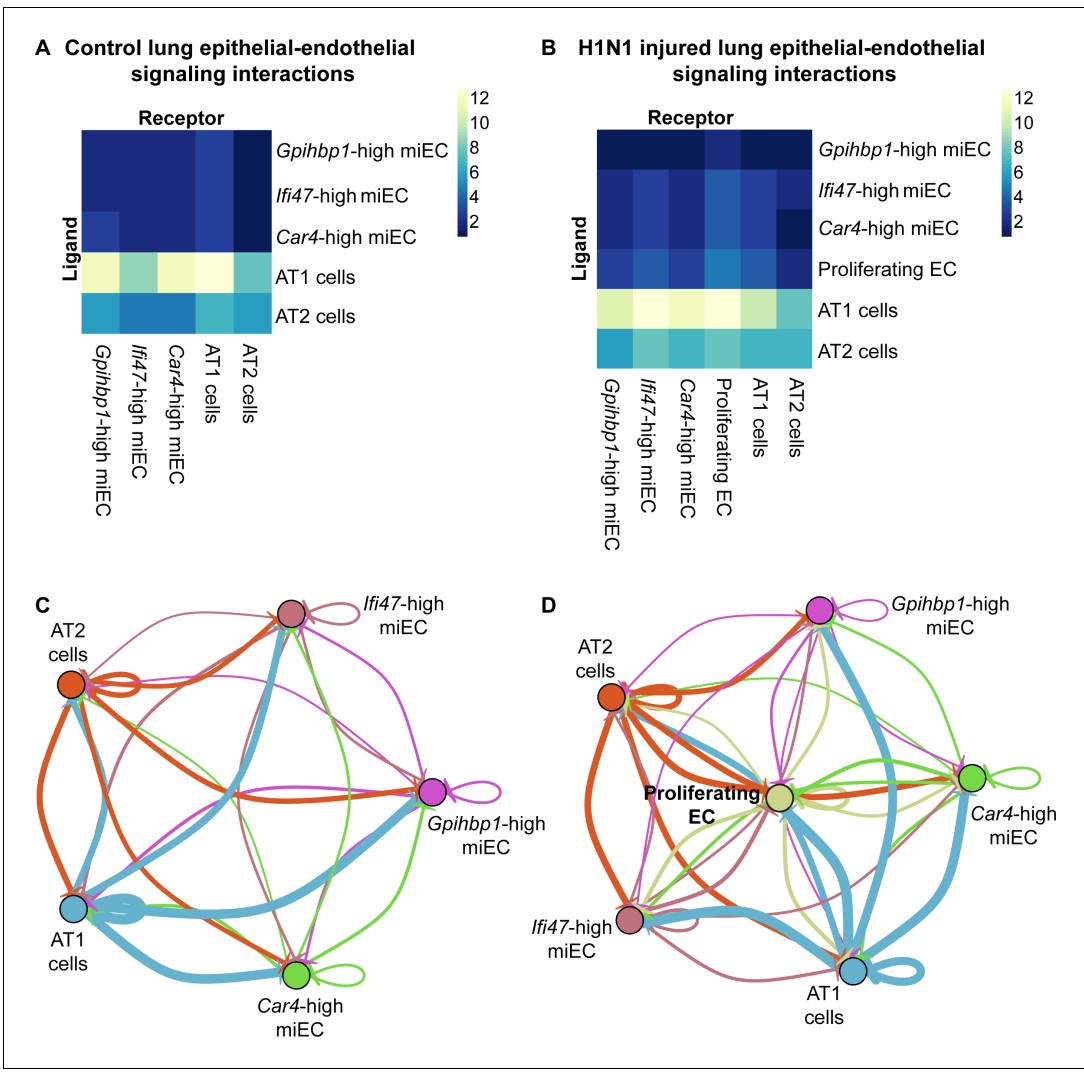

**Figure 7.** *Car4*-high ECs participate in epithelial-endothelial signaling both at homeostasis and after influenza injury. (**A**) Ligand-receptor analysis of scRNA-seq data identifies putative AT1- and AT2-EC signaling axes in the adult mouse lung at homeostasis. *Car4*-high ECs and miECs are predicted to receive signals from both AT1 and AT2 cells, with the strongest possible interactions identified between AT1 cells and ECs. (**B**) Ligand-receptor analysis at 14 dpi demonstrates increased potential for interactions between ECs and AT1/AT2 cells, with the strongest putative signaling interactions between AT1 cells and *Ifi47*-high miECs as well as between AT1 cells and proliferating ECs. (**C**) Similarly, network analysis of putative signaling interactions between epithelial and EC populations reveals the strongest possible interactions between AT1 cells and both miECs and *Car4*-high ECs at homeostasis. (**D**) After H1N1 influenza injury, both AT1 and AT2 cells are predicted to signal robustly to all EC cell types, with the emergent population of proliferating ECs forming a central node in the signaling network.

of a highly proliferative EC population in the lung following influenza injury. The existence of endothelial subtypes that preferentially contribute to regeneration has been investigated in many organs, but the identity and origin of putative endothelial 'progenitor' cells remains a topic of debate (*Basile and Yoder, 2014*). In the aorta and the liver, endothelial regeneration has been attributed to small subsets of endogenous ECs with enhanced proliferative capacity (*McDonald et al., 2018*; *Wakabayashi et al., 2018*). Others have suggested that circulating bone marrow-derived ECs can be delivered to regions of injury to regenerate the endothelium (*Asahara et al., 1999a*; *Asahara et al., 1997*; *Asahara et al., 1999b*; *Lin et al., 2000*; *Takahashi et al., 1999*), but this hypothesis has also proven controversial (*Purhonen et al., 2008*). In the lung, it remained unclear

**Table 4.** Top 12 predicted ligand-receptor pairs for AT1-*Car4*-high EC signaling at homeostasis and after H1N1 influenza injury.

| Receptor | Ligand | Evidence | Receptor cluster | Ligand cluster |
|---|---|---|---|---|
| Itgb5 | Adam9 | literature supported | Car4-high EC | AT1 |
| Flt1 | Vegfa | literature supported | Car4-high EC | AT1 |
| Kdr | Vegfa | literature supported | Car4-high EC | AT1 |
| Itgb1 | Col4a1 | literature supported | Car4-high EC | AT1 |
| Itgb1 | Npnt | literature supported | Car4-high EC | AT1 |
| Itgb1 | Lamc2 | literature supported | Car4-high EC | AT1 |
| Itgb1 | Adam9 | literature supported | Car4-high EC | AT1 |
| Nrp1 | Sema3b | literature supported | Car4-high EC | AT1 |
| Nrp1 | Vegfa | literature supported | Car4-high EC | AT1 |
| Nrp1 | Sema3a | literature supported | Car4-high EC | AT1 |
| Bmpr2 | Bmp4 | literature supported | Car4-high EC | AT1 |
| Fgfr3 | Fgf1 | literature supported | Car4-high EC | AT1 |

whether all ECs, or only a unique subpopulation of progenitors, could contribute to endothelial injury repair. Our scRNA-seq analysis identifies a proliferating EC population that arises after acute lung injury and suggests that these cells are closely related to a subset of endogenous miECs. In addition, EdU incorporation analysis indicates that both *Car4*-high and other ECs proliferate across the 21 day period following injury, demonstrating that regenerative potential is not restricted to a particular subset of the endothelium.

An increasingly useful mouse model to study alveolar destruction and regeneration has been the administration of a sub-lethal dose of the animalized version of the H1N1 influenza virus (PR8). This injury model has been used recently to show the death and regeneration of the alveolar epithelium, primarily by the expansion of a squamous Krt5[+] epithelial cell (**Kumar et al., 2011**; **Vaughan et al., 2015**) and an alveolar epithelial progenitor cell (AEP) (**Zacharias et al., 2018**). This observed epithelial regeneration is necessary to cover the exposed basement membrane and increase the surface area available for gas exchange. To fully understand the regeneration of the alveolar compartment with respect to the recovery of gas exchange, the behavior of the capillary endothelium must be studied in parallel. While the epithelial cells have been observed as the primary infection site of the influenza virus resulting in their death both in vitro and in vivo (**Short et al., 2014**), there is evidence that the endothelium can also be infected and killed in vitro (**Armstrong et al., 2012**). However, the regeneration of the alveolar capillary plexus following influenza injury and the mechanisms by which gas exchange is re-established have remained incompletely understood. Our data sheds crucial light on the potential of the endothelium to proliferate after injury, a cellular behavior that likely serves to regenerate the pulmonary vasculature. High expression of Car4 in the EC population we have identified also suggests that these cells may play a specialized role in gas exchange in the mouse and that their proliferation after injury may contribute meaningfully to restoring the gas exchange interface in the lung: Car4 has been proposed to play a role in carbon dioxide exchange in the rat lung, where it is more broadly expressed in the microvasculature (**Fleming et al., 1993**). A better understanding of the processes of endothelial proliferation and repair of the gas exchange interface will be a crucial step in developing therapies to alleviate patients of hypoxemia and acute respiratory distress syndrome (ARDS) that can be developed post-influenza infection.

An essential role of the endothelium in regeneration is to reestablish oxygenation of the tissue through restoration of blood circulation. The methods the endothelium uses to repopulate areas lacking in vascularity are classically defined as vasculogenesis, or formation of new vascular structures, and angiogenesis, or expansion of the vascular network through generation of new vessels from existing ones (**Potente and Mäkinen, 2017**). Of the cells that contribute to sprouting angiogenesis, there are two distinct EC phenotypes termed tip cells and stalk cells. Tip cells are leading cells that direct the new vessel growth through sprouting, invasion, and migration; stalk cells follow the tip cells and elongate, proliferate and establish the lumen of the new vessel (**Betz et al., 2016**;

*Potente et al., 2011*). Tip cells are distinguished from stalk cells through increased expression of Vegfr2 and Nrp1, which suppresses stalk cell fate (*Aspalter et al., 2015*; *Simons et al., 2016*), whereas stalk cells are characterized by expression of Notch1 and Vegfr1, a decoy receptor that can sequester Vegf (*Blanco and Gerhardt, 2013*). However, tip cell and stalk cell fates are dynamic during sprouting angiogenesis, with ECs possessing the ability to respond to different environments by adjusting Vegfr levels (*Jakobsson et al., 2010*). During late lung development and maturation, the lung vasculature is also thought to develop *via* intussusceptive angiogenesis, a poorly understood process of vessel splitting that may also contribute to regeneration in the lung (*Gianni-Barrera et al., 2011*). Our transcriptomic analysis of the pulmonary endothelium through scRNA-seq shows that key tip cell genes such as *Vegfr2* and *Nrp1* are enriched in the *Car4*-high EC population at homeostasis and after injury; however, the *Car4*-high ECs also express high levels of stalk cell gene *Vegfr1* (*Tables 1*, *2* and *4*). The *Gpihbp1*-high miEC population shares several reported transcripts that define stalk cells such as *Hes1*, *Hey1* and *Jagged1* (*Moya et al., 2012*). However, following influenza injury, the proliferating pulmonary EC population we have identified is closely transcriptionally associated with the miEC population, and endothelial proliferation is evident both in *Car4*-high and other ECs. This is in line with a recent report indicating that endothelial proliferation and regeneration in large vessels does not require the specification of tip or stalk cells, but is instead initially induced in all cells in close proximity to the site of injury (*McDonald et al., 2018*). However, McDonald and colleagues report a second phase of regeneration in which a smaller subset of proliferating ECs in the aorta continue to generate new cells as wound healing progresses. The gene expression profiles of the cells we have identified and their pseudotime trajectories suggest that a similar phenomenon may occur in the microvasculature of the alveoli during regeneration after influenza injury, with a mechanism that does not require tip or stalk cell specification.

Recently, the existence of ECs primed for proliferation in different vascular beds has gained appreciation (*Taha et al., 2017*). Evidence for such cells in the lung has been derived from cell cloning assays that isolate a subset of cells that are prone to proliferate (*Alvarez et al., 2008*; *Lee et al., 2017*). Alvarez and colleagues showed that the clones derived from the microvascular endothelium that were prone to proliferate, termed resident microvascular endothelial progenitors, were enriched in expression of certain progenitor cell antigens (e.g., CD34, CD309) (*Alvarez et al., 2008*). Studies of these ECs in vitro have benefited from the ability to clonally assess the cells' proliferative capacity, but in vitro EC studies are problematic because of the propensity of ECs to de-differentiate once put into culture (*Lacorre et al., 2004*). Thus, studying such cell populations in vivo is essential to eliminate the confounding factors associated with cell culture studies. Evidence of proliferating ECs has been seen in models of idiopathic pulmonary fibrosis (*Ebina et al., 2004*), partial hepatectomy (*Ding et al., 2010*), and left unilateral pneumonectomy (*Ding et al., 2011*). Our studies suggest the presence of a *Car4*-high/*Cd34*-high EC subpopulation in the lung alveolus and the induction of a highly proliferative EC population after injury, but in contrast to previous studies, our data indicate that these two populations are not identical. Future work will further elucidate the relative contributions of *Car4*-high/*Cd34*-high ECs and other ECs to endothelial proliferation, the contributions of the proliferating endothelium to generation of new vasculature in the alveolar space after injury, and the roles of each cell type in re-establishing gas exchange and in other aspects of lung regeneration.

# Materials and methods

## Animals

All mice were C57BL/6J females age 6–8 weeks old (Jackson Laboratories, Bar Harbor, Maine; strain #000664). All animal procedures were approved by the Institutional Animal Care and Use Committee of the University of Pennsylvania. Mice were housed in groups of up to five animals per cage when possible and supplied with extra enrichment if singly housed.

## Bleomycin and influenza injuries

Pharmaceutical grade bleomycin (Hospira) was non-surgically administered intratracheally to anesthetized mice at a concentration of 2 U/kg while control animals were instilled with PBS. Influenza infection was performed with the PR8-GP33 H1N1 influenza (kindly gifted by Dr. John Wherry) at a titrated dose of 0.3 LD$_{50}$ or 1 LD$_{50}$ diluted in PBS, in a 30 µL volume delivered intranasally. In our

animals, these doses reult in a mild to moderate level of tissue damage with different zones of injury visible as previously reported (*Zacharias et al., 2018*). Control animals were administered intranasal PBS. Recovery from infection was assessed after 7, 14, or 21 days as indicated, at which point tissue was harvested for histology or single cell isolation as described below.

### Histology and immunohistochemistry

At the given time point of tissue analysis, mice were euthanized with a lethal dose of $CO_2$ followed by cervical dislocation. Each mouse was then exsanguinated *via* the inferior vena cava. Upon opening the chest cavity, the lungs were further cleared of blood *via* perfusion of cold PBS through the right ventricle at a pressure of 40 cm of water. The trachea was then cannulated and the lungs inflation-fixed with 2% paraformaldehyde at a pressure of 30 cm of water, for 30 min at room temperature followed by 18 hr at 4°C. The lungs were dehydrated and paraffin embedded for microtome sectioning. Hematoxylin and eosin staining was utilized to assess morphology. Immunohistochemistry was used to recognize the antigens of various ECs, using the following antibodies: CD31 (Rat, HistoBio-Tec DIA-310), CD31 (Rabbit, Thermo Fisher MA5-16337), Vwf (Rabbit, Dako A0082), Vcam1 (Rat, eBioscience 14–1061), Gpihbp1 (Rabbit, Thermo Fisher PA1-16976), Plvap (Rat, BioRad MCA2539T), CD34 (Rat, PharMingen 553731), Car4 (Rat, R and D Systems MAB2414), Ki67 (Rabbit, Abcam ab16667), phospho-histone H3 (pHH3, mouse, Cell Signaling Technologies 9706), Sftpc (Goat, Santa Cruz Biotechnology sc-7706), and Hopx (Mouse, Santa Cruz Biotechnology sc-398703).

### RNAscope analysis

Lung tissue was prepared as described for histology and fixed with 4% paraformaldehyde overnight at 4°C. RNAscope (Advanced Cell Diagnostics) was performed using the Fluorescent Multiplex Reagent Kit v2 (ACD 323100) according to the manufacturer's protocol. The following probes were used in this analysis: mm-Vegfc (ACD 492701), mm-Cxcl12 (ACD 422711), mm-Car4-C3 (ACD 468421-C3), mm-Prc1-C3 (ACD 577121-C3), mm-Top2a (ACD 491221), mm-Gpihbp1-C3 (ACD 540631-C3), mm-Plvap (ACD 440221), mm-Pecam1-C2 (ACD 316721-C2), mm-Sftpc (ACD 314101), and mm-Hopx (ACD 405161).

### Single cell isolation and staining

Following the dissection and washing procedure outlined above, unless otherwise stated, all lobes were dissected from the main-stem bronchus and vasculature. All lobes were then minced with a razor blade, and digested in a mixture of collagenase-I, dispase and DNase as previously described (*Zepp et al., 2017*). Following trituration and filtration, red blood cells were lysed, and single cells were resuspended in a FACS buffer containing 1% FBS and 1 mM EDTA. The following antibodies were used for flow cytometry or sorting: CD45-PE-Cy7 (Thermo Fisher 25-0451-82), CD45-APC (Thermo Fisher 17-0451-83), CD45-PerCP-Cy5.5 (Thermo Fisher 45-0451-82), CD31-PE (Thermo Fisher 12-0311-83), CD31-PE-Cy7 (Thermo Fisher 25-0311-82), CD31-APC (Thermo Fisher 17-0311-82), and CD34-eF450 (Thermo Fisher 48-0341-80). For proliferation analysis, cells that had incorporated EdU were fluorescently labeled using the Click-iT Plus EdU Cell Proliferation Kit (Thermo Fisher).

### Fluorescence-activated cell sorting

To isolate ECs for scRNA-seq, single-cell suspensions were first gated to include small cells, gated on singlets, and gated to eliminate dead cells (DAPI negative cells). From the CD45-negative population, the expression of CD31 was analyzed and cell sorting was performed with the MoFlo Astrios Eq (Beckman Coulter). To isolate CD34-high and CD34-low ECs for gene expression analysis, single-cell suspensions were first gated to include small cells, gated on singlets, and gated to isolate CD45-negative cells. From the CD45-negative population, the expression of CD31 and CD34 were analyzed and cell sorting was performed with the FACSAria Fusion Sorter (BD Biosciences).

### RNA isolation and qRT-PCR

Cells were sorted directly from the cytometer into TRIzol-LS (Thermo Fisher) and RNA was isolated according to the manufacturer's protocol. cDNA was generated using the Superscript IV First-Strand Synthesis System (Thermo Fisher). qRT-PCR was performed using Power SYBR Green 2x Master Mix

(Thermo Fisher) on a QuantStudio 7 Flex system (Applied Biosystems). Data was analyzed using a standard curve method, and all calculations were performed using the Pfaffl method to incorporate primer efficiencies calculated from standard curves. Primer sequences are listed in *Table 5*.

## Single cell RNA sequencing using In-Drop and the GemCode platform

All experiments used C57BL/6J female mice aged 6–8 weeks old. For the endothelial-specific scRNA-seq analysis, single cell suspension was prepared from N = 3 individual animals as described above. DAPI-negative, CD45-negative and CD31-positive cells were sorted to obtain heterogeneous pulmonary endothelium. For the whole-lung scRNA-seq analysis, DAPI-negative, CD45-negative cells were sorted to obtain all non-immune lung cells from N = 1 control animal and N = 2 influenza-injured animals. The cell suspension was loaded onto a GemCode instrument (10X Genomics) as previously described (*Zepp et al., 2017*). Briefly, single-cell barcoded droplets were produced using 10X Single Cell 3' v2 chemistry. Libraries generated were sequenced using the HiSeq Rapid SBS kit. For the heterogeneous pulmonary endothelium samples, the resulting libraries were sequenced across two lanes of an Illumina HiSeq2500 instrument in High-output mode. For the whole lung samples, the resulting libraries were sequenced across a single lane of an Illumina HiSeq2500 instrument in Rapid Run mode. Reads were aligned and gene level unique molecular identifier (UMI) counts were obtained using the Cell Ranger pipeline. t-SNE plots for individual heterogeneous pulmonary endothelial samples were generated from the Cell Ranger pipline. Data from individual animals was aggregated for further analyses, which were performed using the Seurat v2.0 pipeline (*Butler et al., 2018*). Briefly, the R package Seurat (version 2.3.4, http://satijalab.org/seurat/) was used to perform scRNA-seq analysis. Cells were first filtered to have >500 detected genes and less than 5% of total UMIs mapping to the mitochondrial genome. Cell cycle phase score was calculated for each cell using the Seurat function *CellCycleScoring*. Data was scaled to remove unwanted variation from number of genes, percent mitochondrial, cell cycle score reduced by regression. Principal component analysis (PCA) was used to reduce the dataset into a smaller number of components (eigen-genes) while preserving the variation of the entire data set. The number of dimensions used in cluster analysis and dimension reduction procedures was determined using the JackStraw test. Dimension reduction was performed using the uniform manifold approximation protection (UMAP) method (*Becht et al., 2018*). Biological replicates were aggregated using the Canonical Correlation Analysis (CCA) method implemented in Seurat. Seurat was used to create dimension reduction and violin plots. Circos plots and Gene Ontology comparison heatmaps were generated with Metascape (*Tripathi et al., 2015*). All data is available on the gene expression omnibus (GEO) accession GEO: GSE128944.

## Ligand-receptor analysis

Ligand-receptor gene pairs were obtained from the FANTOM5 project (http://fantom.gsc.riken.jp/5/suppl/Ramilowski_et_al_2015/). For a given cluster, a ligand or receptor was considered expressed if 30% of cells had a UMI value of greater than 0. A directed graph was constructed with the nodes as

**Table 5.** Primer sequences.

| Gene | Primer type | Forward sequence (5' to 3') | Reverse sequence (5' to 3') |
|------|-------------|------------------------------|------------------------------|
| Cd34 | qRT-PCR | TGGGTAGCTCTCTGCCTGAT | TGGTAGGAACTGATGGGGAT |
| Car4 | qRT-PCR | CAGCTCCTTCTTGCTCTGCT | CCCCAAGCAACTGCTTCTA |
| Ednrb | qRT-PCR | CCCTAAGGGTCTGCATGCTT | GGCCACTTCTCGTCTCTGC |
| Kdr | qRT-PCR | TGTGCGACCCCAAATTCCAT | ACTGGGCATCATTCCACCAA |
| Cd31 | qRT-PCR | CTGGTGCTCTATGCAAGCCT | AGTTGCTGCCCATTCATCAC |
| Gpihbp1 | qRT-PCR | CACAGCGGAACCGACAAAG | ACTGGCAACAGGTCTGAGTC |
| Plvap | qRT-PCR | CGTCAAGGCCAAGTCGCT | AGGGTTGACTACAGGGAGCC |
| Vwf | qRT-PCR | GTGTAAACGGGCATCTCCTC | CCGTCTTCAGTAGCTGGCAT |
| Gapdh | qRT-PCR | CGTCCCGTAGACAAAATGGT | TTGATGGCAACAATCTCCAC |

clusters and edges as number of ligand-receptor pairs between each cluster. R package igraph was used to make the network plots.

### Pseudotime analysis

Cells were embedded to low dimensional space using the UMAP technique (*Becht et al., 2018*). UMAP was chosen for its ability to preserve global structure of the original data set. A cluster-based minimum spanning tree (MST) was performed using the *getLineages* function in the R package Slingshot (*Street et al., 2018*). Developmental curves along these linages were created using the *getCurves* function. Plots were created using the R ggplot2 package.

### Cell counting

Automated quantification of proliferating cells was performed from large single-optical-plane confocal tile scan images containing tens of thousands of total cells. Machine learning was implemented using ilastik software (*Berg et al., 2019*) for supervised pixel classification to distinguish IHC signal from noise as well as to segment closely associated nuclei. Subsequent object detection and quantification was performed using CellProfiler 3.0 (*McQuin et al., 2018*).

### Statistical analysis

Statistical analysis of scRNA-seq data was performed in R. Statistical analysis of qRT-PCR, flow cytometry, and cell counting data was performed using GraphPad Prism software. To determine if differences between experimental groups were statistically significant, a one-way ANOVA was performed, followed by Holm-Sidak multiple comparison tests. A Brown-Forsythe test was used to assess differences in standard deviation between groups. If variances were significantly different, a nonparametric Kruskal-Wallace test was used in place of ANOVA to determine if differences between groups were statistically significant, followed by Dunn's multiple comparison tests. Graphs are displayed as box-and-whisker plots or as bar plots indicating mean and standard deviation. In all cases, each individual data point is displayed on the graph. Values for N as well as p values are indicated in figure legends.

**Key resources table**

| Reagent type (species) or resource | Designation | Source or reference | Identifiers | Additional information |
|---|---|---|---|---|
| Strain, strain background (*Mus musculus*, male, female) | C57BL/6J | Jackson Laboratory | Stock #000664 | |
| Antibody | anti-CD31 (rat monoclonal) | HistoBioTec | DIA-310; RRID:AB_2631039 | IHC (1:200) |
| Antibody | anti-CD31 (rabbit monoclonal) | Thermo Fisher | MA5-16337; RRID:AB_2537856 | IHC (1:50) |
| Antibody | anti-Vwf (rabbit polyclonal) | Dako | A0082; RRID:AB_2315602 | IHC (1:500) |
| Antibody | anti-Vcam1 (CD106) (rat monoclonal) | eBioscience | 14–1061; RRID:AB_467419 | IHC (1:100) |
| Antibody | anti-Gpihbp1 (rabbit polyclonal) | Thermo Fisher | PA1-16976; RRID:AB_2294825 | IHC (1:100) |
| Antibody | anti-Plvap (rat monoclonal) | Bio-Rad | MCA2539T; RRID:AB_1102821 | IHC (1:50) |
| Antibody | anti-CD34 (rat monoclonal) | PharMingen | 553731; RRID:AB_395015 | IHC (1:25) |
| Antibody | anti-Car4 (rat monoclonal) | R and D Systems | MAB2414; RRID:AB_10718416 | IHC (1:100) |
| Antibody | anti-Sftpc (goat polyclonal) | Santa Cruz Biotechnology | sc-7706; RRID:AB_2185507 | IHC (1:50) |

*Continued on next page*

*Continued*

| Reagent type (species) or resource | Designation | Source or reference | Identifiers | Additional information |
|---|---|---|---|---|
| Antibody | anti-Hopx (mouse monoclonal) | Santa Cruz Biotechnology | sc-398703; RRID:AB_2687966 | IHC (1:100) |
| Antibody | anti-Ki67 (rabbit monoclonal) | Abcam | ab16667; RRID:AB_302459 | IHC (1:50) |
| Antibody | anti-phospho-histone H3 (mouse monoclonal) | Cell Signaling Technology | 9706; RRID:AB_331748 | IHC (1:200) |
| Antibody | CD45-PE-Cy7 (rat monoclonal) | Thermo Fisher | 25-0451-82; RRID:AB_2734986 | Flow cytometry (1:200) |
| Antibody | CD45-APC (rat monoclonal) | Thermo Fisher | 17-0451-83; RRID:AB_469393 | Flow cytometry (1:200) |
| Antibody | CD45-PerCP-Cy5.5 (rat monoclonal) | Thermo Fisher | 45-0451-82; RRID:AB_1107002 | Flow cytometry (1:200) |
| Antibody | CD31-PE (rat monoclonal) | Thermo Fisher | 12-0311-83; RRID:AB_465633 | Flow cytometry (1:200) |
| Antibody | CD31-PE-Cy7 (rat monoclonal) | Thermo Fisher | 25-0311-82; RRID:AB_2716949 | Flow cytometry (1:200) |
| Antibody | CD31-APC (rat monoclonal) | Thermo Fisher | 17-0311-82; RRID:AB_657735 | Flow cytometry (1:200) |
| Antibody | CD34-eF450 (rat monoclonal) | Thermo Fisher | 48-0341-80; RRID:AB_2043838 | Flow cytometry (1:50) |
| Chemical compound, drug | PR8-GP33 H1N1 influenza virus | PMID:23516357 | | Dr. E. John Wherry |
| Chemical compound, drug | Pharmaceutical-grade bleomycin | Hospira | Prescription only | 15U/vial |
| Commercial assay or kit | Click-iT Plus EdU Cell Proliferation Kit | Thermo Fisher | C10632 | |
| Other | RNAscope probe mm-Vegfc | Advanced Cell Diagnostics | 492701 | |
| Other | RNAscope probe mm-Cxcl12 | Advanced Cell Diagnostics | 422711 | |
| Other | RNAscope probe mm-Car4-C3 | Advanced Cell Diagnostics | 468421-C3 | |
| Other | RNAscope probe mm-Prc1-C3 | Advanced Cell Diagnostics | 577121-C3 | |
| Other | RNAscope probe mm-Top2a | Advanced Cell Diagnostics | 491221 | |
| Other | RNAscope probe mm-Gpihbp1-C3 | Advanced Cell Diagnostics | 540631-C3 | |
| Other | RNAscope probe mm-Plvap | Advanced Cell Diagnostics | 440221 | |
| Other | RNAscope probe mm-Pecam1-C2 | Advanced Cell Diagnostics | 316721-C2 | |
| Other | RNAscope probe mm-Sftpc | Advanced Cell Diagnostics | 314101 | |
| Other | RNAscope probe mm-Hopx | Advanced Cell Diagnostics | 405161 | |

## Acknowledgements

The authors acknowledge the members of the Children's Hospital of Philadelphia (CHOP) flow cytometry core, the Penn flow cytometry core, the CHOP Center for Advanced Genomics, the Penn Cell and Developmental Biology Microscopy Core, and the Penn Cardiovascular Institute histology

core for their assistance with key experiments reported in this manuscript. We also thank the other members of the Morrisey lab for assistance with experimental techniques and helpful discussion throughout the preparation of this manuscript. The Morrisey Laboratory is supported by grant funding from the National Institutes of Health and the Longfonds Foundation of the Netherlands.

# Additional information

## Competing interests

Edward E Morrisey: Reviewing editor, *eLife*. The other authors declare that no competing interests exist.

## Funding

| Funder | Grant reference number | Author |
|---|---|---|
| National Institutes of Health | R01-HL087825 | Edward E Morrisey |
| National Institutes of Health | U01-HL134745-01 | Edward E Morrisey |
| National Institutes of Health | R01-HL132999 | Edward E Morrisey |
| National Institutes of Health | R01-HL132349 | Edward E Morrisey |
| National Institutes of Health | T32-HL7586-34 | Terren K Niethamer |

The funders had no role in study design, data collection and interpretation, or the decision to submit the work for publication.

## Author contributions

Terren K Niethamer, Conceptualization, Resources, Formal analysis, Supervision, Funding acquisition, Validation, Investigation, Visualization, Project administration; Collin T Stabler, Conceptualization, Validation, Investigation, Visualization; John P Leach, Formal analysis, Investigation, Visualization, Methodology; Jarod A Zepp, Investigation, Visualization; Michael P Morley, Conceptualization, Resources, Data curation, Software, Formal analysis, Validation, Investigation, Visualization, Methodology; Apoorva Babu, Resources, Data curation, Software, Formal analysis, Validation, Visualization, Methodology; Su Zhou, Resources, Data curation, Software, Formal analysis, Investigation, Visualization, Methodology; Edward E Morrisey, Conceptualization, Resources, Supervision, Funding acquisition, Investigation, Project administration

## Author ORCIDs

Terren K Niethamer (ID) https://orcid.org/0000-0002-0914-994X
Edward E Morrisey (ID) https://orcid.org/0000-0001-5785-1939

## Ethics

Animal experimentation: This study was performed in accordance with the recommendations in the Guide for the Care and Use of Laboratory Animals and under the oversight of the Institutional Animal Care and Use Committee (IACUC) of the University of Pennsylvania. All mouse experiments were approved by IACUC under protocol #806345.

## Decision letter and Author response

Decision letter https://doi.org/10.7554/eLife.53072.sa1
Author response https://doi.org/10.7554/eLife.53072.sa2

# Additional files

## Supplementary files

- Transparent reporting form

## Data availability

Single-cell RNA sequencing datasets have been deposited in GEO under accession code GSE128944.

The following dataset was generated:

| Author(s) | Year | Dataset title | Dataset URL | Database and Identifier |
|---|---|---|---|---|
| Niethamer TK, Stabler CT, Morley MP, Babu A, Morrisey EE | 2020 | Defining the role of pulmonary endothelial cell heterogeneity in the response to acute lung injury | https://www.ncbi.nlm.nih.gov/geo/query/acc.cgi?acc=GSE128944 | NCBI Gene Expression Omnibus, GSE128944 |

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
