## [Decision Letter]

**Acceptance summary:**

The reviewers appreciate the novelty and scientific value of mapping the heterogeneity of pulmonary endothelial cells, identification of different subpopulations of these cells, and insights into their roles. Of particular interest is the identification of a new subpopulation of pulmonary endothelial cells that are involved in lung regeneration following injury.

**Decision letter after peer review:**

[Editors’ note: the authors submitted for reconsideration following the decision after peer review. What follows is the decision letter after the first round of review.]

Thank you for submitting your work entitled "Defining the role of pulmonary endothelial cell heterogeneity in the response to acute lung injury" for consideration by *eLife*. Your article has been reviewed by three peer reviewers, and the evaluation has been overseen by a Reviewing Editor and a Senior Editor. The reviewers have opted to remain anonymous.

Our decision has been reached after consultation between the reviewers. Based on these discussions and the individual reviews below, we regret to inform you that your work will not be considered for publication in *eLife* in its present state.

While the reviewers were enthusiastic about the novelty and importance of your study, they also raised substantial concerns that would require extensive additional experimentation for a responsive revision. We would therefore encourage resubmission of your manuscript following revisions, with a commitment to solicit opinions from the original reviewers in order to expedite the re-review process.

We see the strength of the paper in the new insights into the heterogeneity of endothelial cells before and after lung injury, obtained by analysis of single cell RNA sequencing data in the murine lung. Three classes of endothelial cells that were identified and termed "macrovascular", "microvascular" and "regulatory" are of great interest. In particular, the "regulatory" endothelial cells are a novel cell type that is found in alveolar areas, characterized with high expression of CD34 and CAR4, and shown to expand following injury. The reviewers felt that these findings advance current understanding of endothelial cells in the lung, and the dynamic changes during lung injury and repair. The main weaknesses identified by the reviewers include the lack of functional data, and of statistical validation/ analysis (except for the single cell RNA-seq data). Please consult the individual reviews for more detail.

Reviewer #1:

In this study, the authors used single-cell RNA-seq and the latest bioinformatic algorithms to dissect the heterogeneity of endothelial cells in the adult lung, both during homeostasis, as well as during flu or bleomycin-induced injury. They identified three major populations, including REC, a subpopulation of the alveolar endothelium that expresses CD34. They found that CD34^+^ REC cells expanded following injury. Furthermore, following injury, they identified a new population, PEC, which is highly proliferative. Through pseudo-time analysis, they linked PEC to the miEC population, which is distinct from REC. These findings greatly advance current understanding of endothelial cell composition in lung and offer the first glimpse of the dynamic changes contributing to repair. This study will be of great interest to the general readership of *eLife*. Addressing the following comments will enhance the rigor and impact of the study.

1) At the start of the paper and in Abstract, endothelial cells were dissected into three major groups, maEC, miEC and REC. In Figure 4, for uninjured, endothelial cells were separated into more clusters (two for maEC, two for miEC). These additional insights into heterogeneity should be highlighted and further explored by key marker identification and spatial localization (see below).

2) Additional spatial analysis of subpopulations will be very informative. For example, how are the miEC and REC spatially distinct from each other? Where do the *Ifi47*-high vs *Gpihbp1*-high subclusters of miEC localize to? Where are PECs?

3) Following injury, there is an increase of REC, and an appearance of PEC. Localization of PEC in relationship to REC will allow an estimation of the extent of contribution of each population to injury.

4) Aside from the focus on PEC and cell cycle markers after injury, how the transcriptome of the other populations change will be informative, especially in the aspect of signaling.

5) CD34 increase should be quantified by qRT.

6) Parallel to the expansion of REC, how maEC and miEC subclusters change spatially following injury will be informative.

7) Figure 6—figure supplements 1 and 2, provide a better explanation how the different gene plots illustrate the curves.

Reviewer #2:

In this manuscript, the authors present a thorough analysis of scRNA-seq data generated from the endothelial compartment of the lung, before and after injury with influenza and bleomycin. They identify a novel endothelial cell type characterized by high expression of CD34 and of CAR4 that is present in alveolar areas and increases after injury. A role for these cells, called regulatory endothelial cells (RECs), in injury repair is inferred from these findings. The strength of this paper is the thorough mining of the scRNA-seq data. A serious weakness however is that mechanistic and functional conclusions are drawn based on mere inference from the scRNA-seq data, and therefore are speculation at best. Hence, this manuscript is merely descriptive and based on a very limited number of experiments that were maximally mined.

Specific comments:

1) The scRNA-seq studies used purified cells based on CD31 expression. Given the heterogeneity in endothelial phenotypes, does this strategy capture all endothelial cells? Following influenza infection, scRNA-seq was performed on whole lung. Were exactly the same EC populations identified in the control of that experiment, where CD31 purification was not used?

2) After injury, CD34-high ECs increase in the alveolar regions adjacent to regions of severe injury (Figure 3). It is inferred that this represents an accumulation REC, but this might as well simply represent upregulation of CD34 on other ECs. Using other markers for RECs, such as Car4, might be useful. Furthermore, increased CD34 expression is only evaluated by qualitative assessment of IF images. Could this be quantified by flow cytometry? Similarly, the increased in the abundance of these cells is shown in one IF image but is never quantified.

3) A role for proliferating ECs that might be derived from miECs is suggested in lung regeneration. Where were these located?

4) Rigorous quantification is also lacking in the data on cell cycling in Figure 5. Based on scRNA expression, cell cycle status of various populations was assessed. Quite remarkably, these data would suggest that even in steady-state, a very large fraction of all lung cell types are in cell cycle. This is surprising as the lung has been reported previously to be very quiescent, and sheds doubt on the reliability of this type of analysis to assess cell cycle status. It is then stated that after injury, in particular ECs cycle, but this appears true in steady-state as well. Furthermore, there are no statistics to support this conclusion. The “proliferating ECs” of course cycle even more, since they were identified as such by expression of proliferation markers.

5) The EdU incorporation data in Figure 5 also lack controls and statistics. The fraction of S-phase ECs seems remarkably high for steady-state, no data are provided on subpopulations of ECs (CD34-high, for example), no data are provided on other cell types in the sample, no statistical analysis is given, and a staining of PBS-treated mice (i.e. no EdU administration) is lacking as a control for EdU staining.

6) In Figure 6, lineage relations are inferred from scRNA-seq analysis, but the conclusions, while possible correct, are speculation at this point.

7) A similar comment is true for Figure 7, where ligand/receptor pair analysis suggests interactions between ECs and ATI cells. This is very likely correct, but not surprising, given the close apposition of these two cell types. This, however, does not inform on the role of either proliferating ECs or REC in the lung regeneration.

Reviewer #3:

In this manuscript Niethamer et al. report their findings regarding endothelial cell heterogeneity in normal adult and injured mouse lung as defined by single cell RNA-sequencing. Three classes of EC's are defined under homeostatic conditions including macrovascular EC's, microvascular EC's and a third group that the authors refer to as "regulatory" EC's. Parenchymal injury resulted in the appearance of a fourth cluster referred to as proliferative EC's. These findings are important to the field of lung regeneration and endothelial cell biology, but there are a number of technical and conceptual concerns as outlined below.

Specific issues/concerns:

1) Since normal endothelial single cell RNA-Seq data represent aggregated data from multiple individual mice it would be important to show the distinct contributions of these individuals to the entire data set for assessment of batch variability between individuals.

2) Even though the appearance of proliferative EC's in response to acute lung injury is reasonable and well supported by gene expression data, the use of the term regulatory EC's is speculative and not well justified based upon the data. Additional concerns include: a) One of the top differentially expressed genes appearing in REC's, Car4 or carboxypeptidase 4, has been described as broadly expressed within rat microvasculature EC's (Fleming et al., 1993). As such, it is not clear whether REC's represent a novel endothelial cell type, whether they represent a subset of microvascular ECs and/or whether there are species differences that account for this observation; b) none of the marker genes evaluated (with the exception of CD31) are endothelial cell-specific, which makes interpretation of immunofluorescence data challenging.

3) It would be helpful if data generated from uninjured adult mice vs either bleo or influenza virus injured lungs could be combined to determine how EC subsets change with injury. EC clusters generated with the combined transcriptomes of all lung cell types are most likely quite different to those generated if only EC's are evaluated in isolation. Furthermore, it is not clear (without lineage tracing and/or data aggregation) that direct relationships exist between cell types observed in the uninjured vs injured lungs. Accordingly, the authors interpretation of cellular relationships presented in the manuscript may represent an overly simplified scenario and could be somewhat misleading.

4) Reference is made to additional subsets of EC's revealed within single cell preparations of lung tissue after influenza virus infection, Figure 4—figure supplement 3, including lymphatic EC's, venous maEC's and Arterial maEC's. It is not clear how these EC subsets relate to those recovered from normal uninjured lung tissue. Are they lost following selection for CD31^+^ EC's? Candidate receptor-ligand interactions are identified between REC's and ATI cells. However, the basis for selecting these interacting cell types and the absence of validation make this analysis and interpretation rather speculative.

[Editors’ note: further revisions were suggested prior to acceptance, as described below.]

Thank you for submitting your article "Defining the role of pulmonary endothelial cell heterogeneity in the response to acute lung injury" for consideration by *eLife*. Your article has been reviewed by three peer reviewers, and the evaluation has been overseen by a Reviewing Editor and Didier Stainier as the Senior Editor. The reviewers have opted to remain anonymous.

The reviewers have discussed the reviews with one another and the Reviewing Editor has drafted this decision to help you prepare the final submission.

While the reviewers agreed that the revisions of the manuscript are significant and that the revised manuscript is much improved, a few comments remained that would require your attention:

1) Please establish to what extent *Car4^+^* corresponds to CD34-high ECs. For example, by sorting out CD34-high ECs (not just CD34 positive) and showing that Car4 is selectively expressed in this population.

2) Please determine the exact location of the Car4 population with respect to alveolar epithelial cells and other ECs, using Car4, not just CD34, as a marker.

3) Please perform EdU incorporation with a shorter pulse to avoid cytotoxic effect of EdU, which may affect proliferation.

4) Please clarify why is such a high fraction of both ECs and epithelial cells cycling in steady-state according to this analysis in Figure 5—figure supplement 5, and how these data relate to the other inferences from scRNAseq?

---

## [Author Response]

[Editors’ note: the authors resubmitted a revised version of the paper for consideration. What follows is the authors’ response to the first round of review.]

Reviewer #1:1) At the start of the paper and in Abstract, endothelial cells were dissected into three major groups, maEC, miEC and REC. In Figure 4, for uninjured, endothelial cells were separated into more clusters (two for maEC, two for miEC). These additional insights into heterogeneity should be highlighted and further explored by key marker identification and spatial localization (see below).

We agree that it is important to identify key markers and determine the spatial relationships of endothelial cell subsets in the distal lung. We have therefore performed RNAscope analysis using markers of each EC population in the adult lung, both at homeostasis and at 14 days post H1N1 injury. We found that miECs, marked by expression of *Gpihbp1*, *Pecam1*, and *Plvap*, localize to the alveolar capillary endothelium both at homeostasis and after influenza injury, while cells in large vessels do not express *Gpihbp1*. Arterial and venous maECs, marked by *Cxcl12* and *Vegfc*, respectively, localize to large vessels both before and after injury. We have quantified relative numbers of *Car4-high* ECs at homeostasis and after H1N1 injury and determined that while they are localized throughout the capillary space at homeostasis, *Car4-high* ECs are increased in zones of mild to moderate as well as severe injury in H1N1-infected animals. We have added additional supplemental figures containing this analysis (*Figure 3—figure supplements 2-3*).

2) Additional spatial analysis of subpopulations will be very informative. For example, how are the miEC and REC spatially distinct from each other? Where do the Ifi47-high vs Gpihbp1-high subclusters of miEC localize to? Where are PECs?

We have addressed the localization of miECs, proliferating ECs, and *Car4-high* ECs at homeostasis and after injury using RNAscope. We show that miECs and *Car4-high* ECs are transcriptionally rather than spatially distinct, as both are distributed throughout the alveolar space at homeostasis and during repair. Proliferating ECs are rare at homeostasis but localize both to zones of mild to moderate tissue damage and to zones of more severe tissue damage after injury (*Figure 3—figure supplements 2-3; Figure 5—figure supplement 4*).

3) Following injury, there is an increase of REC, and an appearance of PEC. Localization of PEC in relationship to REC will allow an estimation of the extent of contribution of each population to injury.

We have addressed the localization of proliferating ECs and *Car4-high* ECs after H1N1 injury using RNAscope and find that both cell types are localized to zones 1/2 of mild to moderate injury as well as to zone 3 of more severe tissue damage, with an increase in numbers of each cell type after injury with respect to control (*Figure 3—figure supplement 2; Figure 5—figure supplement 4*).

4) Aside from the focus on PEC and cell cycle markers after injury, how the transcriptome of the other populations change will be informative, especially in the aspect of signaling.

To determine how heterogeneous EC populations change with injury, we have integrated the control and H1N1 datasets and performed additional analysis to directly compare cell clusters and their differential gene expression. We discovered that *Car4-high* ECs do not change in expression of key genes that differentiate this population from other EC populations after injury, but we do see statistically significant differences in expression of several signaling genes, genes that regulate the cytoskeleton, and the transcription factor *Hes1*. We have added violin plots to show these gene expression differences in *Car4-high* ECs and gene ontology analysis to describe genes differentially expressed across EC populations from uninjured and injured lungs. We find that after H1N1 injury, ECs upregulate genes related to translation, cell proliferation and survival, as well as cell death. This analysis is shown in two additional supplemental figures (*Figure 4—figure supplements 5 and 6*).

5) CD34 increase should be quantified by qRT.

We have used FACS to isolate *CD34-high* and CD34-low EC populations and have shown that relative *Cd34*, *Car4*, *Ednrb*, and *Kdr* expression is increased in *CD34-high* ECs by qRT-PCR. However, relative expression of other miEC and pan-EC markers such as *Gpihbp1*, *Plvap*, and *Pecam1* is not significantly different between *CD34-high* and CD34-low ECs. We have added an additional supplemental figure containing this data (*Figure 1—figure supplement 3*). We have also quantified changes in number of *CD34-high* cells between control and H1N1-infected mice using flow cytometry and showed that the proportion of *CD34-high* ECs increases after H1N1 injury. This data forms part of the new *Figure 5.*

6) Parallel to the expansion of REC, how maEC and miEC subclusters change spatially following injury will be informative.

We have also addressed this with our RNAscope analysis. Please see response to comment 1 above and the new *Figure 3—figure supplement 3*.

7) Figure 6—figure supplements 1 and 2, provide a better explanation how the different gene plots illustrate the curves.

The feature plots associated with trajectory or curve 1 of the pseudotime analysis (*Figure 6—figure supplement 1*) and curve 2 (*Figure 6—figure supplement 2*) highlight the top differentially expressed genes that define each trajectory. We have refined this analysis to highlight the top 15 genes for each curve, which demonstrate that curve 1 is defined by expression differences between *Car4-high* ECs and proliferating ECs, whereas curve 2 is defined by differences between *Car4-high* ECs and miECs. We have added additional explanation to the main text to clarify these figures – please see paragraph four of subsection “Endothelial Cell Proliferation Increases Significantly During H1N1 Injury Response” and associated figure legends.

Reviewer #2:1) The scRNA-seq studies used purified cells based on CD31 expression. Given the heterogeneity in endothelial phenotypes, does this strategy capture all endothelial cells? Following influenza infection, scRNA-seq was performed on whole lung. Were exactly the same EC populations identified in the control of that experiment, where CD31 purification was not used?

As expected, we identified additional endothelial cell populations in the whole lung scRNA-seq experiments. These include separation of venous and arterial macrovascular endothelium and lymphatic endothelium. The additional separation of endothelial cell populations is likely the result of differences in unbiased clustering due to an increased number of cell types and additional variation in gene expression between cell types analyzed in the whole-lung experiments as well as a less biased cell isolation procedure. We have added sentences to the main text to clarify these differences and have better justified our choice of CD31 expression to purify endothelial cells.

2) After injury, CD34-high ECs increase in the alveolar regions adjacent to regions of severe injury (Figure 3). It is inferred that this represents an accumulation REC, but this might as well simply represent upregulation of CD34 on other ECs. Using other markers for RECs, such as Car4, might be useful. Furthermore, increased CD34 expression is only evaluated by qualitative assessment of IF images. Could this be quantified by flow cytometry? Similarly, the increased in the abundance of these cells is shown in one IF image but is never quantified.

We have quantified the increase in *CD34-high* ECs after H1N1 infection by flow cytometry and have found that the proportion of *CD34-high* ECs does increase significantly with influenza injury. We have added this quantification and associated statistical analysis to the new *Figure 5*. In addition, we have quantified numbers of *Car4*-expressing cells in different zones of injury following H1N1 infection by RNAscope and have seen a corresponding increase in these cells in regenerating regions, indicating that our IHC data likely does not represent simply an upregulation of CD34 protein. This quantitative analysis of *Car4*-expressing cells is shown in an additional supplemental figure (*Figure 3—figure supplement 2*).

3) A role for proliferating ECs that might be derived from miECs is suggested in lung regeneration. Where were these located?

We have addressed PEC localization after H1N1 influenza infection with RNAscope analysis and found that these cells localize to both regions of mild to moderate injury and regions of more severe injury. Please see response to reviewer 1, comments 2 and 3 above, as well as the new Figure 5—figure supplement 4.

4) Rigorous quantification is also lacking in the data on cell cycling in Figure 5. Based on scRNA expression, cell cycle status of various populations was assessed. Quite remarkably, these data would suggest that even in steady-state, a very large fraction of all lung cell types are in cell cycle. This is surprising as the lung has been reported previously to be very quiescent, and sheds doubt on the reliability of this type of analysis to assess cell cycle status. It is then stated that after injury, in particular ECs cycle, but this appears true in steady-state as well. Furthermore, there are no statistics to support this conclusion. The “proliferating ECs” of course cycle even more, since they were identified as such by expression of proliferation markers.

To assess and quantify changes in endothelial cell populations and their proliferation after H1N1 influenza injury, we have performed flow cytometry experiments at 7, 14, and 21 dpi and quantified the relative abundance of CD31^+^/ *CD34-high* and CD31^+^/ CD34-low cell populations and their EdU incorporation at homeostasis and during repair. We found that the proportion of CD31^+^/ *CD34-high* ECs increases over the course of the injury response, and that both *CD34-high* and CD34-low ECs incorporate EdU during regeneration. We have added an additional figure to the manuscript that includes this data (the new *Figure 5 and associated Figure 5—figure supplements 1-3). We have also clarified the methods of the cell cycling analysis of our scRNA-seq datasets and have added text to the manuscript to emphasize the predictive nature of this analysis, as well as its advantages and limitations for assessing cell-cycle status. Please see paragraph three of subsection “Endothelial Cell Proliferation Increases Significantly During H1N1 Injury Response” and Figure 5—figure supplement 5.*

5) The EdU incorporation data in Figure 5 also lack controls and statistics. The fraction of S-phase ECs seems remarkably high for steady-state, no data are provided on subpopulations of ECs (CD34-high, for example), no data are provided on other cell types in the sample, no statistical analysis is given, and a staining of PBS-treated mice (i.e. no EdU administration) is lacking as a control for EdU staining.

We have performed additional and more rigorous EdU incorporation experiments in control and H1N1-infected mice described in the new Figure 5 and associated Figure 5—figure supplements 1-3. We have also added a control experiment to assess the reliability of the EdU water administration and fluorescent labeling strategy we employed. We found that there is a small amount of background fluorescence in single cells of H1N1-infected mice not given EdU water but subjected to the fluorescent labeling assay. This may result in a slightly larger percentage of S-phase ECs quantified at both homeostasis and after injury but is very minor in comparison to the difference we observed in EdU incorporation between control and flu-infected mice (Figure 5—figure supplement 1). We have also analyzed proliferation in *CD34-high* and CD34-low ECs and found that both populations incorporate EdU over the course of injury repair. In analyzing this quantitative proliferation data, we have used statistical analysis to support our conclusions, which is shown in the new *Figure 5.* In addition, we analyzed proliferating non-ECs by RNAscope (*Top2a^+^/Prc1^+^* cells that are *Pecam1^-^*) and found that like proliferating ECs, these cells are increased in both zones 1/2 and zone 3 after H1N1 injury (*Figure 5—figure supplement 4*).

6) In Figure 6, lineage relations are inferred from scRNA-seq analysis, but the conclusions, while possible correct, are speculation at this point.

While we agree that these analyses are exploratory rather than confirmatory, we would argue that mechanistic analysis of these predicted relationships is beyond the scope of the current manuscript and forms the basis for future work. We have edited the text to clarify the role of these analyses in our conclusions as a whole and have modified our language to emphasize the speculative nature of these analyses.

7) A similar comment is true for Figure 7, where ligand/receptor pair analysis suggests interactions between ECs and ATI cells. This is very likely correct, but not surprising, given the close apposition of these two cell types. This, however, does not inform on the role of either proliferating ECs or REC in the lung regeneration.

Similar to Figure 6, the analyses presented in Figure 7 are exploratory rather than confirmatory. We have modified our language in the text to emphasize the fact that this analysis suggests predicted interactions and to clarify the role of this analysis in the manuscript as a whole.

Reviewer #3:1) Since normal endothelial single cell RNA-Seq data represent aggregated data from multiple individual mice it would be important to show the distinct contributions of these individuals to the entire data set for assessment of batch variability between individuals.

Both our CD31^+^ endothelial cell-specific and our H1N1 injury whole-lung scRNA-seq datasets contain aggregated data from multiple individual mice. We have therefore added additional supplemental figures showing the overlay of data from individual mice in each integrated dataset. The EC-specific dataset analyzed in Figure 1 is aggregated from three individual mice; in the new Figure 1—figure supplement 2, we show that all seven clusters 0-6 are identified in the dataset for each individual mouse. The H1N1 whole-lung dataset analyzed in Figure 4 is aggregated from two individual mice; in the new Figure 4—figure supplement 5, we find that integration of these two datasets with the control whole-lung dataset results in similar clustering and cell populations as does individual analysis. These figures nicely demonstrate the concordance between individual datasets used for aggregation.

2) Even though the appearance of proliferative EC's in response to acute lung injury is reasonable and well supported by gene expression data, the use of the term regulatory EC's is speculative and not well justified based upon the data. Additional concerns include: a) One of the top differentially expressed genes appearing in REC's, Car4 or carboxypeptidase 4, has been described as broadly expressed within rat microvasculature EC's (Fleming et al., 1993). As such, it is not clear whether REC's represent a novel endothelial cell type, whether they represent a subset of microvascular EC's and/or whether there are species differences that account for this observation; b) none of the marker genes evaluated (with the exception of CD31) are endothelial cell-specific, which makes interpretation of immunofluorescence data challenging.

We agree that our use of the term “regulatory endothelial cell” is a potential overstatement, and we have therefore revised the text to refer to these cells as “*Car4-high* ECs” in all text and figures. We have also performed further analysis of gene expression in *Car4-high* ECs by qRT-PCR of FACS-isolated *CD34-high* and CD34-low ECs and have found that *CD34-high* (*Car4-high*) ECs express higher levels of *Cd34*, *Car4*, *Ednrb*, and *Kdr* than CD34-low ECs, but similar levels of *Pecam1*, *Gpihbp1*, *Plvap*, and *Vwf* (*Figure 1—figure supplement 3*). We have also assessed the spatial localization of *Car4-high* ECs by RNAscope; similar to our CD34 IHC analysis, RNAscope for *Car4* indicates that these cells are localized throughout the alveolar space at homeostasis (Figure 3—figure supplement 2). These data have led us to conclude that these cells represent a novel miEC population. We agree that high Car4 expression may not define a subpopulation of ECs in all species and have revised the Discussion to include reference to the above article. In our RNAscope analysis (please see response to reviewer 1, comment 1 above), we have used co-labeling with probes that detect pan-endothelial gene expression to provide more rigorous analysis of ECs from each subpopulation in control and H1N1-infected conditions (Figure 3—figure supplements 2-3).

3) It would be helpful if data generated from uninjured adult mice vs either bleo or influenza virus injured lungs could be combined to determine how EC subsets change with injury. EC clusters generated with the combined transcriptomes of all lung cell types are most likely quite different to those generated if only EC's are evaluated in isolation. Furthermore, it is not clear (without lineage tracing and/or data aggregation) that direct relationships exist between cell types observed in the uninjured vs injured lungs. Accordingly, the authors interpretation of cellular relationships presented in the manuscript may represent an overly simplified scenario and could be somewhat misleading.

We have added additional analysis of our scRNA-seq data to compare gene expression within EC subtypes before and after injury, including integration of control and H1N1 datasets (please see response to reviewer 1, comment 4 above). We have also added additional text to clarify the differences observed in our endothelial cell-specific and whole-lung scRNA-seq datasets (please see response to reviewer 2, comment 1 above).

4). Reference is made to additional subsets of EC's revealed within single cell preparations of lung tissue after influenza virus infection, Figure 4—figure supplement 3, including lymphatic EC's, venous maEC's and Arterial maEC's. It is not clear how these EC subsets relate to those recovered from normal uninjured lung tissue. Are they lost following selection for CD31+ EC's? Candidate receptor-ligand interactions are identified between REC's and ATI cells. However, the basis for selecting these interacting cell types and the absence of validation make this analysis and interpretation rather speculative.

We have added additional text to the manuscript to clarify these points (please see responses to reviewer 2, comment 1 and reviewer 2, comment 7 above).

[Editors’ note: what follows is the authors’ response to the second round of review.]

1) Please establish to what extent Car4+ corresponds to CD34-high ECs. For example, by sorting out CD34-high ECs (not just CD34 positive) and showing that Car4 is selectively expressed in this population.

We assessed this using the FACS strategy outlined in Figure 1—figure supplement 3, panel A, where our gating strategy separates *CD34-high* cells (right panel, upper right-hand box) and CD34-low/CD34^-^ cells (right panel, upper left-hand box). qRT-PCR comparing these two populations reveals that *CD34-high* cells express a significantly higher level of *Car4* than CD34-lowcells (>6 fold) (panel C). Of note, these data were included in the previous version of the manuscript.

2) Please determine the exact location of the Car4 population with respect to alveolar epithelial cells and other ECs, using Car4, not just CD34, as a marker.

To address this question, we have conducted RNAscope for *Car4*, *Pecam1* as a marker for all ECs, and either *Hopx* (alveolar type 1 cells) or *Sftpc* (alveolar type 2 cells). Spatial analysis of these images shows that both *Car4-high* ECs and other *Pecam1*^+^ ECs can be found juxtaposed with both AT2 and AT1 cells. Using these methods, we do not observe a preferential localization of *Car4^+^* endothelial cells to AT1 or AT2 cells. We have added a new figure (Figure 3—figure supplement 3) as well as text to the manuscript describing these results.

3) Please perform EdU incorporation with a shorter pulse to avoid cytotoxic effect of EdU, which may affect proliferation.

To directly address the issue of EdU toxicity, we have compared endothelial cell proliferation as well as total proliferation by IHC using phosphohistone H3 (pHH3) in EdU-treated and untreated animals from the same H1N1 injury experiment at a defined time point of 14 dpi. We feel that this analysis addresses the issue of EdU toxicity directly, whereas redoing the entire experiment with a shorter EdU pulse would not completely address this (such studies would not have detected any possible toxicity) and would extend the revisions well past the 2-month time frame. Moreover, our choice of concentration (0.2 g/L) and timeline (7 days) forin vivoEdU administration in drinking water has plenty of precedent in the literature [1-2], and indeed, EdU administration in drinking water has been performed at up to fivefold higher concentration [3] as well as for longer periods up to 90 days with no reported toxicity [4]. Our new results indicate that 1) EdU treatment does not affect proliferation in the lung using the dose/timing in our experiments and 2) Total proliferation, EC proliferation, and proliferation in *Car4-high* ECs are all increased at 14 days post H1N1 infection as measured by IHC for pHH3 in control and injured animals, confirming our previous results. We have added a new figure (Figure 5—figure supplement 1) and text to the manuscript describing this new data. Of note, to complete this analysis, we have implemented a cell counting pipeline that utilizes the ilastic [5] and CellProfiler [6] programs to accurately and rigorously segment nuclei and assess cell numbers expressing multiple marker genes. Please refer to the Materials and methods section for a more detailed description of this new tool.

4) Please clarify why is such a high fraction of both ECs and epithelial cells cycling in steady-state according to this analysis in Figure 5—figure supplement 5, and how these data relate to the other inferences from scRNAseq?

Figure 5—figure supplement 6 (with new numbering resulting from addition of the supplemental figure described in (3) above) contains an informatic derivation of cell cycle state based on expression of genes related to cell cycle progression. This is therefore a predictive analysis rather than a functional one and does not allow us to determine whether a cell is actively cycling. Our subsequent EdU incorporation, RNAscope, and IHC analysis all confirm that the distal lung is quiescent at steady-state, with very little proliferation in endothelial cells or elsewhere. Therefore, although informatic analysis indicates that cells express some genes associated with S and G2/M phases at homeostasis, functional data indicates that most of these cells are not actively cycling. The discrepancy may relate to differences between cells poised to re-enter the cell cycle versus cells actively cycling. We have added text to the manuscript to reflect this. However, if the editors and reviewers feel that these data are confusing, we would be happy to remove them from the manuscript. The data contained in Figure 5 and its other supplements all provide more convincing functional evidence.

References

1) Meln et al.,*Molecular Metabolism*(2019) PMC: 6531874

2) Kryvalap et al.*JBC*(2016) PMC: 4697161

3) Amano et al.,*Cell Metabolism*(2013) PMC: 3931314

4) Clarke LE et al.,*J Neurosci*(2012) PMC: 3378033

5) Berg et al., *Nat Methods* (2019) PMID: 31570887

6) McQuin et al., *PLoS Biol* (2018) PMC: 6029841